# Suprachoroidal Injection: A Novel Approach for Targeted Drug Delivery

**DOI:** 10.3390/ph16091241

**Published:** 2023-09-01

**Authors:** Kevin Y. Wu, Jamie K. Fujioka, Tara Gholamian, Marian Zaharia, Simon D. Tran

**Affiliations:** 1Department of Surgery, Division of Ophthalmology, University of Sherbrooke, Sherbrooke, QC J1G 2E8, Canada; yang.wu@usherbrooke.ca (K.Y.W.);; 2Faculty of Medicine, Queen’s University, Kingston, ON K7L 3N6, Canada; 3Faculty of Medicine, University of Ottawa, Ottawa, ON K1H 8M5, Canada; 4Faculty of Dental Medicine and Oral Health Sciences, McGill University, Montreal, QC H3A 1G1, Canada

**Keywords:** ophthalmology, ocular diseases, drug delivery, controlled drug release, retina, posterior segment diseases, ocular drug bioavailability, suprachoroidal injection

## Abstract

Treating posterior segment and retinal diseases poses challenges due to the complex structures in the eye that act as robust barriers, limiting medication delivery and bioavailability. This necessitates frequent dosing, typically via eye drops or intravitreal injections, to manage diseases, often leading to side effects with long-term use. Suprachoroidal injection is a novel approach for targeted drug delivery to the posterior segment. The suprachoroidal space is the region between the sclera and the choroid and provides a potential route for minimally invasive medication delivery. Through a more targeted delivery to the posterior segment, this method offers advantages over other routes of administration, such as higher drug concentrations, increased bioavailability, and prolonged duration of action. Additionally, this approach minimizes the risk of corticosteroid-related adverse events such as cataracts and intraocular pressure elevation via compartmentalization. This review focuses on preclinical and clinical studies published between 2019 and 2023, highlighting the potential of suprachoroidal injection in treating a variety of posterior segment diseases. However, to fully harness its potential, more research is needed to address current challenges and limitations, such as the need for technological advancements, refinement of injection techniques, and consideration of cost and accessibility factors. Future studies exploring its use in conjunction with biotech products, gene therapies, and cell-based therapies can lead to personalized treatments that can revolutionize the field of ophthalmology.

## 1. Introduction

The landscape of ocular drug delivery is in constant evolution, presenting new challenges and opportunities in the field of ophthalmology. Treating posterior segment and retinal diseases is particularly challenging due to the eye’s complex structures that act as barriers to drug delivery and bioavailability [1]. Traditional administration methods, such as eye drops, periocular and intravitreal (IV) injections, and systemic medications, often require frequent dosing and can result in substantial side effects with long-term use [2]. Recently, suprachoroidal (SC) injection has emerged as a novel strategy for targeted drug delivery to the posterior segment of the eye, offering an innovative approach to address these challenges [3].

The suprachoroidal space (SCS), an anatomical niche nestled between the sclera and the choroid, provides a minimally invasive conduit for precise medication delivery. This approach not only enhances drug concentrations in the posterior segment, increasing drug bioavailability and duration of action but also minimizes the risk of corticosteroid-related adverse events through compartmentalization [4]. The potential of SC injection has been highlighted by promising results from recent preclinical and clinical studies.

This review offers a comprehensive overview of SC injection, covering its rationale, techniques, biomechanics, and implications in treating diverse ocular diseases, particularly those affecting the posterior segment. We also explore the current challenges and future prospects of this technique.

While prior review articles have mainly addressed the use of SC injection for macular edema secondary to conditions such as uveitis, diabetic retinopathy, or CRVO [5,6,7], our review uniquely extends beyond existing clinical data. We explore not only the application of this technique in clinical settings but also delve into preclinical studies for other ocular conditions such as glaucoma, retinitis pigmentosa, and various chorioretinal diseases. This review goes beyond simply informing clinicians about existing indications as we shed light on new therapeutic possibilities emerging from preclinical studies yet to be applied to human subjects. Furthermore, our examination of the biomechanics of SC injection serves to bridge the gap between theoretical understanding and clinical practice by exploring how alterations in various physical parameters of the injection can influence its applicability.

To achieve this, we performed an extensive literature review, mainly focusing on articles published post-2020, to ensure the inclusion of the latest advancements and insights. Through this exploration, we aspire to capture the current state of this technique, elucidate potential avenues for improvement and innovation, and provide a reference point for further research and clinical applications in this rapidly evolving field.

## 2. Anatomy and Physiology

### 2.1. Choroid

The choroid, a layer in the eye, is nourished by blood from the posterior ciliary arteries. This blood flows through two key sub-layers of choroidal vessels, the Haller and Sattler layers, to reach the choriocapillaris, where arterial pressure reduces to a lower level. The choroid’s thickness varies across its expanse, being thickest in the central macular region and thinnest at the ora serrata. Typically, in most 50-year-olds, the choroid measures about 287 μm subfoveally, though thickness can vary with age and ocular disease conditions [8].

Located in the posterior pole, the choriocapillaries feature an intricate capillary network that is more irregular toward the periphery. The choroid’s composition also includes loose connective tissue, fibroblasts, and melanocytes. In the post-choriocapillaris, blood is gathered in venules, followed by larger channels, to drain into the superior and inferior ophthalmic veins through the vortex veins [9].

The choroid plays a crucial role in supplying nutrients, particularly oxygen, to the retina, one of the body’s most metabolically active tissues (Figure 1). Approximately 90% of the oxygen consumed by the outer retina, housing photoreceptors, and retinal pigment epithelium (RPE), is provided by choroidal circulation. Furthermore, the choroid features the highest blood flow rate of any tissue in the body. Despite the high metabolic demand of these tissues, the exiting venous blood maintains high oxygen tension, reflecting the choroid’s efficient function in nutrient delivery and metabolic waste removal [9].

### 2.2. Sclera

The sclera consists of collagen and a small number of elastic fibers embedded in a proteoglycan matrix. Its thickness varies, being the thinnest near the muscle insertion sites and thicker posterior to the limbus, where it terminates [10].

An essential characteristic of the sclera is its permeability, facilitating bidirectional molecular transport. Its permeability enables drug delivery via injections into the sub-Tenon space. However, the sclera’s hydrophilic nature means that its permeability to hydrophobic or amphiphilic substances, including certain medications, can vary. This property is a crucial factor to consider when planning periocular injections of pharmacological agents [11].

The thickness of the human sclera varies between individuals and across different regions of the eye. A histomorphometric study conducted by Vurgese, Panda-Jonas, and Jonas on 238 human eyes found that in non-axially elongated eyes (axial length ≤ 26 mm, with the average axial length usually being around 23 mm), the sclera was thickest at the posterior pole (0.94 ± 0.18 mm), followed by the peri-optic nerve region (0.86 ± 0.21 mm), and the midpoint between the posterior pole and equator (0.65 ± 0.15 mm). The thickness decreases toward the limbus (0.50 ± 0.11 mm), the ora serrata (0.43 ± 0.14 mm), and the equator (0.42 ± 0.15 mm) and is the thinnest at the peripapillary scleral flange (0.39 ± 0.09 mm) [11]. The relatively small inter-individual variability in scleral thickness supports a standardized approach to injections, allowing clinicians to use a uniform microneedle length. For most cases, a 0.9 mm microneedle suffices, while certain scenarios may require a slightly longer 1.1 mm microneedle [12,13]. This finding simplifies the suprachoroidal injection procedure, as it reduces the need for individualized microneedle length adjustments based on patient-specific ocular characteristics.

For axially elongated eyes (i.e., myopic eyes, axial length > 26 mm), scleral thinning is more pronounced at and posterior to the equator, with a greater thinning as it nears the posterior pole [11]. Interestingly, the inter-individual variability in scleral thickness is found to be different in high myopia. However, despite this regional scleral thinning, it does not necessitate alterations in the needle length for suprachoroidal injections. This is due to the fact that these injections are typically administered at an anterior location where the thickness of the sclera is relatively consistent among individuals [14].

### 2.3. Suprachoroidal Space

The SCS is a potential area nestled between the sclera and the choroid (Figure 2) [15]. This space is often in close contact due to the intraocular pressure (IOP) [16] and the presence of attaching fibers [17]. However, the introduction of fluid, whether internally or externally, can transform this potential space into a more defined one.

The SCS has shown considerable expansion following the injection of certain drugs in this area. A study involving the injection of triamcinolone acetonide in the SCS demonstrated a notable increase in mean SCS width, from 9.9 μm to 75.1 μm [4]. This expansion proved the influence of SC injection in manipulating the SCS’s physical attributes. However, the increase was temporary, with the SCS width returning to approximately 14.9 μm a month after the final injection, revealing no lasting impact on the SCS’s anatomy [4].

The SCS, located between the sclera and choroid, has boundaries that are anatomically distinct. Anteriorly, the SCS extends up to the scleral spur, a pivotal landmark that marks the juncture of scleral attachment to the ciliary body. Posteriorly, the SCS is situated near the optic nerve and short posterior ciliary arteries [3,12,18,19]. It is essential to recognize the anatomical placement of the SCS when considering pharmacological interventions, such as SC injections. Distinguishing the SCS from the subretinal space is crucial, as the former lacks the immune privilege characteristic due to its position outside the blood–retinal barrier. For clarity, it is worth revisiting the structure of the blood–ocular barrier. This barrier consists of the vascular endothelium of the retina, which is non-fenestrated and bound by tight junctions. Although the choroidal vessels are fenestrated, the barrier is maintained through the presence of tight junctions within the retinal pigment epithelium (RPE) [3,12,18,19].

## 3. Route of Administration

Numerous routes are available for administering ocular medications, each with unique strengths and weaknesses. Standard methods include systemic delivery (e.g., oral, intravenous, and subcutaneous routes) and local delivery methods (e.g., topical eye drops, periocular or IV injections, and IV implants). While these methods can be effective, they can also come with certain limitations [20]. An overview of the advantages and disadvantages associated with each ocular drug administration method is summarized in Table 1: Comparison of Different Ocular Drug Administration Methods and Figure 3 [2,21].

### 3.1. Topical Administration

Topical administration, often in the form of eye drops, is a prevalent non-invasive method for ocular drug delivery. However, it is associated with several challenges as a consequence of the anatomy and physiology of the eye.

First, the concentration gradient from the tear reservoir to the cornea or conjunctiva drives passive absorption, but only approximately 20% of a drop (about 10 μL of the 50 μL drop) is retained in the eye [25]. Within 3–4 min, half of the administered medication has typically left the eye, with a turnover rate of roughly 15% per minute. Factors such as reflex tearing, consecutive dosing, and the small cul-de-sac of the eye contribute to a fast tear turnover time, further accelerating drug clearance and challenging the effective drug absorption [2,21].

Second, medications need to travel through the dual barriers posed by the hydrophobic tight junctions formed by the epithelium and endothelium, as well as the hydrophilic stroma layer of the cornea (Figure 3) [26]. The inherent low permeability of the cornea and sclera impedes this process, diminishing the bioavailability of the topically administered drug.

Due to the relatively impermeable corneal barriers and high tear turnover rates, topical administration often necessitates frequent, high-dose applications. This approach can cause local and systemic side effects, potentially reducing patient compliance [27]. Remarkably, studies have indicated that the rate of medication non-compliance in the general population is approximately 80% [28]. Such challenges are often exacerbated in certain populations, such as the elderly and those with physical disabilities.

Additionally, the exposure of unaffected tissue to drugs may lead to certain side effects. For instance, chronic usage of topical steroids can result in complications such as cataracts and ocular hypertension [22]. Similarly, topical prostaglandins can lead to undesirable periocular aesthetic concerns [29].

Overall, while topical application serves as a primary mode of ocular drug delivery, these complexities underline the need for advancements in methods of drug delivery.

### 3.2. Systemic Administration

Oral delivery has been explored as a potential drug administration route for ocular conditions, either standalone or in combination with topical delivery [30,31,32,33]. Although it could be a noninvasive, patient-preferred method for managing chronic retinal diseases, the limitations of oral administration include its reduced accessibility to many targeted ocular tissues, necessitating high doses for therapeutic efficacy. However, high dosage can result in systemic side effects, making safety and toxicity critical considerations [34,35].

For the oral route to be effective in ocular applications, high oral bioavailability is a key requirement. Furthermore, following oral absorption, molecules must navigate through systemic circulation and across the blood–ocular barriers, notably the blood–aqueous and blood–retinal barriers (Figure 4). The blood–retinal barrier is further stratified into an inner barrier, protected by the fenestrated endothelium of retinal vasculature, and an outer barrier, upheld by tight junctions within the RPE. The functional properties and inherent barriers posed by these protective ocular structures represent significant challenges for the systemic drug administration [34,35].

For instance, systemic medications, such as steroidal and nonsteroidal anti-inflammatory drugs and biologic and nonbiologic immunomodulatory agents, can effectively treat uveitic macular edema (UME) but are often recommended for bilateral disease or cases resistant to local therapy due to AEs such as infections and GI disturbances [20,36,37]. Furthermore, nonsteroidal anti-inflammatory drugs and systemic immunomodulatory agents may increase the risk of GI disturbances when used alone or combined with steroids [20,36,37].

### 3.3. Periocular Injection

Periocular drug administration is employed to address the inefficiencies of topical and systemic dosing, both of which struggle to deliver therapeutic drug concentrations to the posterior segment [2]. The periocular route, including subconjunctival, subtenon, retrobulbar, and peribulbar administrations, is comparatively less invasive than IV drug administration [2].

Subconjunctival injections can improve water-soluble drug absorption by bypassing the conjunctival epithelial barrier. However, drug access to the posterior eye segment is still restricted due to various barriers, including dynamic ones such as conjunctival blood and lymphatic circulation [38,39,40]. These dynamic barriers often result in rapid drug elimination, reducing ocular bioavailability and vitreous drug levels post-administration [38,39,40]. While the permeable sclera allows for some molecules to reach the neural retina and photoreceptor cells [20,41], the choroid’s high blood flow can remove a significant drug fraction before it reaches its target. Further limitations are posed by the blood–retinal barriers formed by the tight junction within the RPE, restricting drug availability to the photoreceptor cells.

### 3.4. Intravitreal Injection

In the realm of ocular drug delivery, IV administration offers numerous advantages and has been widely adopted as a first-line therapy for conditions such as neovascular age-related macular degeneration (nAMD) [42], diabetic macular edema (DME) [43], and macular edema secondary to retinal vein occlusion (RVO) [44]. It has been well-accepted due to its proven safety and efficacy alongside the convenience of application in an office setting. Its popularity arises from its salient advantages, such as direct medication delivery to the retina and vitreous by bypassing corneal and scleral barriers (as compared to topical eyedrops) and the ability to circumvent the blood–retinal barrier (unlike systemic medications). This ensures high bioavailability in the target area, triggering rapid therapeutic effects. This method also addresses issues with patient non-compliance typically associated with topical eyedrops, as the administration is under the direct control of an ophthalmologist. The versatility of IV administration covers a spectrum of therapeutics, including anti-VEGF agents and corticosteroids, rendering it highly effective for a wide range of retinal diseases [45].

However, IV injections are not without certain drawbacks and potential complications. Severe complications can occur, which include the risk of endophthalmitis, retinal detachment, and vitreous hemorrhage. Furthermore, IV steroids specifically have associated complications such as increased intraocular pressure and cataract development [43]. These complexities not only challenge the treatment process but also hinder achieving optimal visual outcomes [43]. Minor side effects and inconveniences, such as floaters post-injection and the potential for systemic absorption and resultant side effects, can also adversely affect patient satisfaction and treatment adherence [43].

Another hindrance to optimal IV injection outcomes lies in the necessity for a frequent injection regimen due to the short half-life of these drugs [46,47,48]. After the IV injection, the drug is primarily expelled either anteriorly or posteriorly. Anterior elimination entails drug diffusion through the vitreous to the aqueous humor and then removal via aqueous turnover and uveal blood flow. Posterior elimination involves the drug permeating the blood–retinal barrier, necessitating either effective passive permeability or active transport mechanisms. Consequently, compounds with hydrophilic properties and large molecular weights tend to have longer half-lives within the vitreous humor [20]. In contrast, hydrophobic drugs with smaller molecular weights tend to have a shorter half-life, implying the need for frequent injections. Regular in-office visits can be burdensome for individuals residing in rural areas or managing chronic conditions, leading to patient non-compliance, which inevitably reduces the overall effectiveness of the treatment [48]. Furthermore, the clearance rate can vary based on patient-specific factors, such as age and whether the patient has undergone a vitrectomy, which adds another layer of complexity to the treatment regimen [49].

To address the challenges of short treatment duration and frequent in-office visits, intraocular implants have been strategically designed. For instance, the Multicenter Uveitis Steroid Treatment (MUST) randomized controlled trial (RCT) (NCT00132691) evaluated the efficacy and safety of a 0.59 mg fluocinolone acetonide (FA) intraocular implant, which releases the drug over approximately 30 months [50]. The study found that the FA implant improved uveitic inflammation control and reduced macular edema (ME) more effectively than systemic corticosteroids in the short term, although the differences diminished by 24 months [51]. Additionally, the FA implant was linked to a fourfold increase in the risk of elevated intraocular pressure (IOP) requiring intervention [51]. After seven years of extended follow-up, patients who received systemic therapy demonstrated better visual acuity than those with IV FA implants [51].

In the field of gene therapy, while early trials on an anti-VEGF transgene product (Adverum Biotechnologies) have shown promise, concerns arise due to the significant inflammatory responses of injecting it intravitreally [52,53,54,55]. The vitreous poses an additional hurdle for retinal gene delivery due to its component, particularly hyaluronan, which can interact with cationic lipid, polymeric, and liposomal DNA complexes, leading to severe aggregation and immobilization of DNA/liposome complexes [56,57]. Moreover, the inner limiting membrane (ILM), which separates the retina and vitreous, serves as a barrier to the retinal delivery of gene-based therapies [58]. For choroidal diseases, drug transport from the vitreous to the choroid is difficult due to the presence of the RPE, which serves as a barrier (i.e., outer blood–retinal barrier formed by the tight junction within RPE) [9].

Emerging alternatives, such as subretinal and SC drug delivery, could potentially offer longer-lasting effects, reducing injection frequency and limiting side effects, including gene therapy-induced inflammation [59].

### 3.5. Subretinal Injection

Subretinal delivery presents a compelling avenue for retinal gene therapy, especially for the treatment of retinal degeneration and vascular diseases. This approach involves the direct introduction of viral vectors into the subretinal space—an immune-privileged site—thus allowing targeted treatment for the RPE and outer retina while reducing the likelihood of immune reactions [24].

The first FDA-approved gene therapy for RPE65-associated inherited retinal dystrophy, Voretigene neparvovec-rzyl (Luxturna), has provided promising outcomes [60,61]. The potential of gene therapy also extends to conditions such as diabetic retinopathy (DR) and age-related macular degeneration (AMD), suggesting the possibility of a single-dose treatment for these chronic diseases. Early data from studies using subretinal adenoviral vector anti-VEGF gene therapy point toward a significant decrease in treatment burden and an encouraging safety profile for nAMD [62].

However, it is important to note that subretinal delivery does have its own set of challenges. It is invasive in nature, requiring a vitrectomy for administration. Moreover, the localized nature of the injectate can limit its distribution within the subretinal space, potentially confining the therapeutic effects to the area surrounding the injection site [59].

## 4. Suprachoroidal Injection: Rationale

SC injection offers a treatment pathway that is both minimally invasive and potentially long-lasting, effectively combining the advantages of IV and subretinal injections [59,63]. Notably, the SCS can be accessed using a variety of tools, including catheters, needles, and microneedles. The use of a microneedle provides more precise targeting and control during in-office deliveries to the SCS compared to traditional hypodermic needles [59].

### 4.1. Advantages over the Intravitreal Injection

SC injection stands out as a method that enables precise and targeted delivery to the retina, RPE, and choroid. By bypassing barriers such as the ILM and vitreous, which are commonly encountered in the IV drug administration [64], this method achieves broader bioavailability across the diseased retina and choroid [59,63].

The unique compartmentalization provided by SC injection within the SCS plays a pivotal role in its advantages. This containment restricts drug exposure to target tissues, minimizing unnecessary contact with the anterior segment [59,64], which, in turn, reduces the risk of complications such as cataract formation and elevated intraocular pressure [24]. Furthermore, this compartmentalization minimizes systemic absorption, leading to fewer systemic side effects [24]. Supporting these benefits, a 2022 study involving rabbits demonstrated that SC delivery of TRIESENCE provided a 12-fold greater exposure to the RPE, choroid, sclera, and retina compared to IV delivery [24]. Remarkably, the same study found that SC delivery resulted in a 460-, 34-, and 22-fold reduction in drug exposure to anterior chamber structures, specifically the lens, iris ciliary body, and the vitreous humor, respectively [24]. This decreased exposure highlights the enhanced safety profile of SC drug delivery [24].

Moreover, SC injection offers a sustained-release mechanism, reducing the frequency of injections and, consequently, the number of patient appointments [59]. Unlike the IV space, the SCS is not immune-privileged, thereby theoretically posing a lower risk of endophthalmitis, although further studies are required to substantiate this claim [24]. Furthermore, by avoiding injections directly into the vitreous cavity, risks associated with this method, such as traumatic cataracts and retinal tears with their subsequent potential for detachments, might also be diminished. Further enhancing the patient experience, SC injection mitigates the risk of visual axis obstruction, leading to fewer incidences of post-injection floaters, a common side effect with IV methods [59].

### 4.2. Advantages over Subretinal Injection

When compared to subretinal injection, the SC method can be administered in an outpatient setting, reducing the need for complex surgical procedures such as vitrectomy [59,63]. Furthermore, it offers the potential to provide a broader distribution of drugs across the posterior segment [64].

### 4.3. Drug Suspension Size and Formulation Viscosity

Current research is exploring the potential to alter drug suspension size and formulation viscosity in order to adjust the duration and distribution of the injected drugs. This flexibility could allow precise tailoring of drug delivery, ensuring that the right amount of medication reaches the target location.

### 4.4. Cost-Effectiveness

Over a 10-year horizon, a simulated US adult patient-level model evaluated the cost-effectiveness of suprachoroidal triamcinolone acetonide (SC-TA) compared to the best supportive care for UME derived from the PEACHTREE trial. The authors determined that, at willingness-to-pay thresholds of $50,000 or more (2020 US dollars) per quality-adjusted life-year gained, SCTA was a cost-effective procedure [5].

The combined practicality, enhanced safety profile, proven efficacy, targeted delivery, and durability offered by SC drug delivery have made SC injections an innovative treatment modality for diverse ocular conditions. This underlines the imperative for more extensive research into this therapeutic strategy, which is also the focus of this review article.

## 5. Suprachoroidal Injection Techniques

Access to the SCS is typically achieved using the following three methods: microcatheters; needles; and microneedles. Catheter-based technology, such as the iTrack microcatheter, involves the insertion of a 250 A microcatheter into the SCS [65]. Using an incision site in the sclera, the microcatheter is carefully advanced through the SCS toward the designated treatment zone. The placement of the microcatheter can be confirmed and adjusted as needed using indirect ophthalmoscopy to ensure accurate positioning. Advantages of this technique include precise targeting and visualization, as the catheter can be guided with a flashing diode [7]. However, drawbacks of using this method include the fact that it is an invasive procedure that typically requires an operating room, and the success of the injection relies on the skills of the administrator. As with all procedures, there are risks of adverse events and complications such as vitreous penetration, SC hemorrhage, choroidal tears, irregularities in choroidal blood flow, post-operative inflammation, scleral ectasia, retinal detachment, wound abscess, and endophthalmitis among others.

Injection into the SCS can also be achieved by a free-hand technique using a standard hypodermic needle attached to a Hamilton syringe or insulin syringe [7,66,67]. In this approach, the needle is inserted through the sclera behind the limbus, with or without sclerotomy. Slow and controlled advancement of the needle is performed by applying gentle pressure on the plunger, and the injection is administered gradually upon experiencing a loss of resistance. The use of standard hypodermic needles offers the advantage of readily available materials and a less invasive procedure, making it more accessible and convenient. However, this technique lacks visualization capabilities and, thus, requires a high level of training and skill to ensure precise injection. There is a risk of inadvertently injecting into unintended structures, which can lead to complications such as choroidal hemorrhage and retinal detachment. The difficulty in controlling the insertion depth and angle further increases the likelihood of unintentional IV or subretinal injections.

Hollow microneedles are miniature devices used primarily for transdermal drug delivery [7,13]. These microneedles possess a hollow internal compartment filled with drug dispersion or solution and tips with small holes. Recent advancements in microneedle technology have revolutionized the accessibility of the SCS without the need for surgical procedures. The SCS microinjector is a manual piston syringe used for accessing the SCS non-surgically. It is designed to be used with varying microneedle lengths (900 µm or 1100 µm) depending on the scleral thickness, which is penetrated until a loss of resistance is felt [68]. In order to minimize the risk of vitreous perforation, the microneedle is slightly longer than the scleral and conjunctival layers. Drug administration with the SCS microinjector involves several steps. First, under local anesthesia, a 900 µm microneedle is positioned perpendicularly 4.5 mm posterior to the limbus at the pars plana (the flat area of the ciliary body). Gentle pressure is applied to the ocular surface to create a sealing gasket between the needle hub and conjunctiva, preventing the backflow of the injectate. The injection into the SCS occurs over 5–10 s while maintaining the perpendicular position and compressing the conjunctiva. After the injection, upon reaching the SCS, the needle hub should be kept in place for 3 to 5 s. If scleral resistance is still felt, an 1100 µm microneedle should be used instead [69].

Microneedle technology offers precise control in reaching the SCS, unlike standard hypodermic needles [7,13]. Short microneedles limit penetration into the SCS by penetrating the sclera consistently and facilitating drug delivery to the intended site. Once inside the SCS, the injectate spreads posteriorly and circumferentially, ensuring broad coverage. In contrast to catheter-based procedures, microneedle-based SC injections can be performed in an office setting under aseptic conditions without requiring vitrectomy or sclerotomy. These SCS microneedles are specifically designed to match the approximate thickness of the sclera and offer several advantages, including ease of use, minimal pain, affordability, minimal invasiveness, low training requirements, outpatient suitability, and improved safety profile. As a result, they represent the most promising route for drug administration.

## 6. Biomechanics of Suprachoroidal Injection

### 6.1. Injection Forces

SC injections performed with a microinjector only require the mechanical force applied by a physician’s hand to deliver the therapeutic formulation into the SCS. An average glide force of 2.07 N was recorded as the mechanical pressure necessary to administer suprachoroidal triamcinolone acetonide (SCTA) into the SCS of in vivo porcine eyes [70,71]. An SCTA prototype, X-TA, was specifically formulated by Muya and collaborators for SC injections aiming to reduce friction, minimize foaming, and prevent microbubble formation. The glide force of X-TA was compared to Triesence TA (TRI) formulated for IV injections, as well as water and air, which served as control measurements. Interestingly, they found that the injection of X-TA required a smaller glide force (0.73 N) with lower variability than TRI (1.31 N), closer to that of air (0.19 N) and water (0.23 N). This discrepancy could be attributed to the larger size of TRI particles, which likely contributed to the higher and more inconsistent glide force required for its delivery. Adapting formulations specifically for SC injections has the potential to reduce the required glide force and improve the stability of the procedure. This is important for achieving higher success rates of therapeutic injections into the SCS for the treatment of various ocular diseases [72].

### 6.2. Volume and Injections into Multiple Quadrants

Optimizing SC surface coverage is crucial for the effective treatment of posterior segment pathologies such as AMD, DR, and RVOs. The volume of injectate has a significant role in drug distribution, directly influencing therapeutic coverage and, as a result, outcomes.

In an animal study by Gu and collaborators, it was found that injecting 20 μL of saline and TA expanded the SCS by 130% to 200% more than a 10 μL injection, highlighting the effect of the volume of injectate [73]. However, ex vivo experiments using rabbit eyes demonstrated that larger volumes injected in the SCS primarily increased circumferential coverage rather than thickness [19]. Quantitatively, injecting ≥75 μL of fluorescein covered at least 50% of the choroidal surface, while 100 μL covered approximately 75% of the posterior globe [74,75]. At smaller volumes, thickness expansion appeared to be influenced by the volume of injectate, whereas at larger volumes, circumferential distribution played a more significant role. This discrepancy could be attributed to the presence of lamellae structures between the choroid and the sclera, which restrict expansion of the SCS and direct the flow of fluid posteriorly with larger volumes [75]. One potential solution to overcome this restriction is to degrade the fibrils using collagenase, as adding a 0.5 mg/mL collagenase preparation to the formulation of 1 μm latex microparticles resulted in a 20% to 45% increase in SCS coverage during ex vivo experiments with rabbit eyes [76]. Simultaneous injection of collagenase and latex particles in a single injection yielded better outcomes than subsequent injections. However, this approach is not suitable for individuals with a latex allergy. Non-uniform fluid distribution of injectate in the SCS occurs partially due to anatomical barriers, such as the scleral spur, optic nerve, and short ciliary arteries. To enhance coverage and achieve even more fluid distribution, Nork and collaborators performed injections in multiple eye quadrants in a rabbit study [77]. They were able to demonstrate that injecting 50 μL of sodium fluorescein in the superior-temporal and inferonasal quadrants was sufficient to cover the entire choroid, suggesting that multiple injections in opposing quadrants could be an effective strategy to maximize SCS coverage [77]. This comprehensive coverage of the entire choroid surface can be particularly important in the treatment of certain generalized choroidal–retinal dystrophies, such as retinitis pigmentosa. In these cases, it is beneficial to deliver the therapeutic agent across all affected areas, making the approach of multiple injections in opposing quadrants particularly valuable.

### 6.3. Viscosity and Polymeric Solution Formulations

Viscosity is another modifiable characteristic of formulations that can be adjusted to optimize the treatment of specific posterior segment conditions. The behavior of fluids with different viscosities, such as Hank’s Balanced salt solution (HBSS, viscosity of ≈75,000 ± 35,000 cPs), DisCoVisc (viscosity of ≈75,000 ± 35,000 cPs), and 5% carboxymethyl cellulose (CMC, viscosity of ≈200,000 cPs) has been analyzed [75,78]. The size of the injection site opening was found to be 0.43 ± 0.06 mm for HBSS and 2.1 ± 0.1 mm for 5% CMC. The SCS collapse rate after injection was 19 min for HBSS and 9 days for CMC. HBSS was no longer detectable after 0.33 ± 0.05 days, whereas it took 1.7 ± 0.7 days for the 5% CMC to be cleared. Higher viscosity agents tend to induce greater expansion and slower SCS collapse rates due to their low aqueous solubility and slow dissolution rate [68].

The shear-thinning (S-T) property is the non-Newtonian behavior of fluids, characterized by lower viscosity as the shear rate increases. Kim and collaborators investigated the following fluids with different behaviors: HBSS (lower S-T fluid); DisCoVisc, 2.2 wt% 950 kDa hyaluronic acid (HA) (moderate S-T fluids); and 1.7 wt% 700 kDa CMC and 3 wt% 90 kDa methylcellulose (higher S-T fluids) [79]. These fluids were mixed with fluorescent particles, injected, and analyzed. HBSS spread more rapidly compared to moderate and high S-T fluids. After 14 days, circumferential spread increased more for moderate S-T fluids and remained relatively unchanged for high S-T fluids [79]. In addition to these results, Jung and collaborators developed an in situ-forming hydrogel of Bevacizumab and HA cross-linked with poly (ethylene glycol) diacrylate. When liquid Bevacizumab was used alone, it was cleared from the SCS within 5 days. However, when formulated with a high molecular weight (MW) HA or cross-linked to poly (ethylene glycol) diacrylate, it took 1 month and 6 months, respectively, for clearance [80]. These studies demonstrate that viscosity is an important property of fluids and that polymeric solutions can be manipulated to optimize SC injections. The utilization of polymers extends the duration of drug retention within the suprachoroidal space (SCS) and assists in managing its dispersion. Polymers with elevated molecular weight (MW) and a moderate degree of non-Newtonian behavior, such as hyaluronic acid (HA), have been demonstrated to promote the dispersion of particles. Conversely, polymer solutions with pronounced non-Newtonian characteristics, such as methylcellulose (MC) and carboxymethylcellulose (CMC), tend to remain stationary at the injection site. This approach potentially enhances the efficiency of the treatment by maintaining a higher and more sustained concentration of the drug at the target site, thereby reducing the rate of systemic absorption.

### 6.4. Particle Suspensions

Particle suspensions, which gradually dissolve over time, can be beneficial in achieving prolonged therapeutic effects on ocular tissues. The clearance kinetics of these suspensions are significantly influenced by the MW of the particles. In order to investigate this relationship, Chiang and collaborators conducted experiments using injections of fluorescein and HBSS containing fluorescent polymeric particles. While fluorescein was detectable for only 1 day, the fluorescent particles in the HBSS suspension were detected for up to 21 days post-injection [81]. Similarly, another experiment comparing fluorescein (detectable for 12 h) with fluorescent dextran (higher MW and hydrodynamic radius than fluorescein, detectable up to 4 days) observed that MW influences the duration of detection [82,83].

Particle size is another factor that significantly affects the clearance kinetics of particle suspensions. Clearance routes include diffusion into the sclera and choroid, transscleral leakage, and choroidal blood flow. The fenestrations of the choriocapillaris allow particles with an estimated size range of 6 to 12 nm to be cleared through choriocapillaris circulation. Larger particles ranging from 20 nm to 10 μm were found to remain in the SCS for several months [79]. In an experiment conducted by Hackett and collaborators using polymeric microparticles of poly (lacto-co-glycolic acid) (PLGA) loaded with acriflavine (ACF) of size 7 μm in Brown Norway rats, the particles were present in the SCS throughout the 16-weeks of the study [84].

However, larger particles may encounter injection difficulties due to their potential blockage by collagen fibers within the sclera, especially when using short microneedles. The spacing between collagen fibers is estimated to be around 300 nm, making injections of particles sized 500–1000 nm more challenging. Therefore, glide force becomes an important parameter for successful SCS delivery when injecting formulations of larger particles [85]. Patel and colleagues observed a significant difference in the distribution of small particles (20–100 nm) and large particles (500–1000 nm) with shorter needles. However, for longer microneedles (1000 μm), all particles behaved similarly, suggesting successful reach of the SCS [3].

Manual injections of particle suspensions ranging from 20 nm to 10 μm were performed by Kim and collaborators and Chiang and collaborators, who both reported similar findings, with particles remaining in the SCS for up to 3 months and consistent fluorescence levels for all particle sizes. This indicates that particle size does not substantially affect SCS distribution but can influence clearance kinetics within a specific size range determined by the choriocapillaris and scleral extracellular matrix pore size [19,79]. The results of these two studies indicate that particle suspensions can be adapted to suit therapeutic needs.

### 6.5. Osmotic Characteristics and Ionic Charges of Formulation

Osmotic power and ionic charge are additional factors that appear to impact drug distribution within the SCS. In a study by Jung and collaborators, osmotic power was demonstrated through the use of highly concentrated HA solution injection, following a less concentrated HA solution containing fluorescent particles. The highly concentrated HA solution was able to attract fluid, leading to greater expansion of the SCS and displacement of the fluorescent particles toward the posterior pole [86].

Regarding the influence of ionic charge on drug distribution, negatively charged nanoparticles were injected into the SCS, and their concentration in the posterior pole of the eye increased significantly when exposed to a positively charged cathodal current in the same study [87]. Touchard and collaborators obtained similar results with non-viral negatively charged DNA particles exposed to electric current in a rat model [88]. Hence, osmotic power and ionic charge are fluid properties that can be leveraged to optimize drug spread in SC injections.

### 6.6. Compartmentalization and Duration of Injectates in the SCS

SC injection offers an advantage in terms of injectate compartmentalization and prolonged drug effect, thereby minimizing side effects by avoiding exposure to distal ocular tissues. Microscopic analysis of the compartmentalization of SC injections in porcine eyes with red fluorescent sulforhodamine injectate was conducted by Patel and collaborators [3]. Using in vivo studies on rabbit models, the degree of choroid and retina targeting by SC and IV injections was further quantified. SC injection of fluorescein resulted in the detection of 10 to 100 times more content in the choroid and retina compared to IV injections, where the signals produced were more uniformly distributed throughout the visual axis [12]. Tyagi and collaborators performed SC injections of NaF and found peak concentrations in the choroid retina that were 36 times higher than subconjunctival injections and 25 times higher than IV injections. The SC route provided a 6-fold higher choroid and retina NaF exposure compared to the posterior subconjunctival route and 2-fold higher exposure compared to the IV route [23].

Similar compartmentalization was observed in studies involving TA injections in rabbit models. Negligible amounts of TA were detected in the anterior segment of the eye after 91 days, while the sclera, choroid, and RPE displayed the highest concentrations [89]. SCTA led to scleral, choroidal, and retinal concentrations 12 times higher than IV injections, while concentrations in the lens, iris-ciliary body, and vitreous humor were 460, 32, and 22 folds lower, respectively. Aqueous humor levels were negligible, and plasma levels were undetectable [90]. Similar observations were made with other molecules, such as Axitinib and A01017, showing maximal concentrations in the SCR at 67 days and 90 days, respectively [91,92]. Compartmentalization was further demonstrated in another study on SCTA, which showed plasma levels below 1 ng/L for up to 24 weeks [93].

### 6.7. Tailoring Suprachoroidal Drug Delivery

In summary, SC injections are a valuable strategy for ocular drug delivery. Its effectiveness is dependent on numerous parameters such as injection force, volume of injectate, formulation characteristics, and compartmentalization. A critical factor is viscosity, with higher viscosity agents favoring drug localization more anterior to the ocular equator and lower viscosity enabling greater posterior delivery. Furthermore, higher viscosity formulations slow clearance rates, thereby prolonging the drug’s duration of action. In addition to viscosity, the size of particle suspensions is key [64]. Larger particles tend to stay longer in the SCS and are less subject to washout by the choroidal circulation, thus extending the therapeutic effect [82,83]. By skillfully manipulating these parameters, researchers and clinicians can tailor drug delivery to individual patients’ needs, depending on the specific location and chronicity of the ocular disease being treated [1,2,3]. This personalization could significantly enhance treatment outcomes and patient satisfaction, marking an important step toward precision medicine in ophthalmology.

## 7. Suprachoroidal Injection in Ocular Diseases

This section presents a detailed overview of the application of SC injection in the treatment of various ocular diseases, encompassing a spectrum of studies from preclinical to clinical stages. We encourage readers to consult Table A1 in Appendix A for a more in-depth understanding of each individual study referenced in our analysis.

### 7.1. Macular Edema

#### 7.1.1. Suprachoroidal Injection for Macular Edema Secondary to Non-Infectious Uveitis

Favorable results from preclinical studies, where SCTA injection effectively concentrated corticosteroid levels in the retina, RPE, and choroid while minimizing exposure to anterior chamber structures, paved the way for subsequent clinical trials [5]. The PEACHTREE clinical trial, led by Yeh and colleagues, demonstrated the effectiveness and safety of SCTA in treating ME secondary to non-infectious uveitis (NIU). This randomized, double-masked study encompassed 160 eyes assigned to receive either 4.0 mg SCTA at two time points (day 0 and week 12) or a sham injection in a 3:2 ratio. Remarkably, by week 24, nearly half (46.9%) of the eyes treated with SCTA exhibited significant visual improvement, gaining 15 or more early treatment diabetic retinopathy study (ETDRS) letters from baseline, compared to only 15.6% in the control group. This improvement was observed as early as 4 weeks and was maintained through week 24. Similarly, a difference of more than 100 μm in mean central subfield thickness (CST) was observed between the two groups at week 4 (−148 μm in the SCTA group vs. −4 μm in the control group) and maintained by week 24. Moreover, ME resolution (CST < 300 μm) was substantially higher in the SCTA group (53% vs. 2% in controls) as early as week 4 until the study’s end. SCTA was also effective in reducing the need for rescue therapy (13.5% vs. 72% in controls) and in extending the median time to the first rescue to 89 days versus 36 days. While treatment-related AEs were noted in both groups at rates of 29% (SCTA) and 8% (control), the incidence of cataracts and elevated IOP were comparable with no treatment related serious AEs. Thus, the PEACHTREE trial demonstrated the robust potential of SCTA as a therapeutic approach to managing ocular diseases, showing clinically significant vision improvement and suggesting that further exploration in this domain is both warranted and promising [94].

Henry and collaborators confirmed that patients undergoing SCTA injection for the treatment of ME due to NIU experienced improved visual and anatomical outcomes alongside comparable rates of AEs, regardless of age at 24 weeks [95]. A post-hoc analysis by Merrill and colleagues also found that benefits demonstrated in the PEACHTREE remained consistent regardless of concurrent systemic corticosteroid use or steroid-sparing therapy. The only difference was that among patients receiving steroid therapy, the mean best corrected visual acuity (BCVA) change was statistically significant, while the mean CST change was not statistically significant between the SCTA and control groups. Among patients who did not receive steroid therapy at baseline, 14.7% of those treated with SCTA versus 69.4% in the control group received rescue therapy. In contrast, for patients who received steroid therapy at baseline, the need for rescue therapy was 10.7% versus 80.0% in the SCTA and control groups, respectively. These results showed a statistically significant difference between the groups that received no steroid therapy and those that received it [96]. These post-hoc analyses reaffirm that SCTA outperforms sham treatment in terms of both functional and anatomical outcomes. Notably, the use of SCTA did not result in a statistically significant improvement in CST for patients who received concurrent steroid therapy. This finding highlights the need for additional studies that identify the suitability of SCTA therapy for patients on different concurrent treatment regimens. Certain factors, such as prior steroid therapy, may have the potential to impact treatment outcomes.

Khurana and colleagues conducted MAGNOLIA, an extension safety study of the PEACHTREE trial in 2022. Their results revealed that the statistically significant improvement in BCVA and CST reduction was maintained for the 28 eyes that received treatment versus the 5 eyes in the control group until 48 weeks. While the need for rescue treatment was not statistically different between each group, the median time to rescue therapy was significantly longer in the SCTA group compared to the control group (257 days versus 55.5 days). The proportion of individuals with at least one ocular AE was 64.3% for SCTA eyes and 60% in the control group, with the most common AE being subcapsular cataract. In the 48-week duration study period, eight patients (seven in the SCTA and one in the control group) had a cataract-related AE, with two of them requiring surgery (both in the SCTA group). In total, 14.3% of SCTA patients had at least 1 elevated IOP reading >10 mmHg, versus 0% in the control group, with no one requiring surgical management [97]. In addition to MAGNOLIA, Henry and colleagues (2022) conducted a safety clinical trial called AZALEA. The study involved 38 eyes, with 53% of them having ME, who received two 4.0 mg SCTA injections spaced 12 weeks apart. The results demonstrated that SCTA was well tolerated and safe for a duration of over 24 weeks in these patients. The mean BCVA, CST, and excess retinal thickness improved at all visits until week 24; however, there were no statistical analyses to determine the significance of this improvement. The proportion of individuals with a treatment-related AE was 18.4%, with pain (7.9%), IOP rise >10 mmHg (15.8%), and IOP >30 mmHg (5.3%) being the most common. No one required surgery for their elevated IOP, but 87.5% required treatment with IOP lowering drops. The formation or worsening of a cataract was experienced by 10.5%, with none requiring surgery [93]. As seen in both AZALEA and MAGNOLIA, the efficacy of SCTA treatment for ME due to NIU was maintained for up to 48 weeks. Few experienced treatment-related AEs, most notably, cataract progression and IOP elevation.

Prior to their pivotal clinical trial, PEACHTREE, Yeh and collaborators conducted DOGWOOD, a randomized, masked study in 2019. They concluded that a 4.0 mg injection of SCTA in patients with ME secondary to NIU was efficacious and well-tolerated. A total of 22 eyes received 4.0 mg or 0.8 mg SCTA in a 4:1 ratio and were assessed at 1 and 2 months. In the group that received 4.0 mg, CST and BCVA were both statistically significantly improved at 1 and 2 months. Of the 10 subjects in the 4.0 mg SCTA treatment group with an anterior cell grade >0 cells at baseline, all subjects showed improvement at month 2 with a 60% resolution (change to score of 0). The remaining seven patients had an anterior cell grade of 0, with 85.7% maintaining this grade status at 2 months. Among 10 subjects in the 4.0 mg SCTA treatment arm that had a vitreous haze score >0 at baseline, 80% showed improvement. Patients with a vitreous haze score of 0 at baseline maintained their grade in the second month. At least one AE was reported by 47% and 100% of the 4.0 mg SCTA and 0.8 mg SCTA groups, respectively, with the most common events being eye or injection site pain (18%), conjunctival hemorrhage (13.6%) and ME (13.6%) that all resolved without treatment. No significant elevation in IOP nor serious treatment-related AEs were reported [98]. Hanif and colleagues (2021) found that 4.0 mg SCTA was safe and efficacious for the treatment of ME secondary to NIU, supporting the initiation of larger-scale studies. Their single-arm study involving 30 eyes with ME secondary to NIU found statistically significant differences in CMT and BCVA from baseline to 1 and 3 months. While five patients were found to have lenticular changes, none of these changes had an effect on patient-reported vision, and there was no statistically significant change in IOP at 3 months [99]. Munir and colleagues, using a similar methodology to Hanif and colleagues, found that the mean BCVA improved as early as 1 week for up to 6 months in 50 patients with ME, of which 30 of them were secondary to NIU (other diagnoses included vascular disorders, diabetic ME, sarcoidosis, and pseudophakic edema). In terms of AEs, IOP was highest at 6 months in cases with baseline IOPs of 11–15 mmHg up to 35 mmHg and highest at 1 month in the baseline IOP group of 16–20 mmHg up to 30 mmHg [100].

The functional and anatomic improvement experienced post-SCTA injection for the treatment of ME due to NIU has been repeatedly shown in multiple clinical studies. The improvement in BCVA, CST, and longer time to rescue therapy was present regardless of patient age or concurrent use of systemic corticosteroid or steroid-sparing therapy. The sustained drug effect, as evidenced by patients not requiring rescue treatment for a mean time of 257 days, has the potential to reduce the treatment burden in comparison to current therapeutic regimens. Safety studies, such as MAGNOLIA, confirm that these benefits are longstanding for up to 48 weeks with minimal AEs.

Due to the unique compartmentalization and ocular distribution provided by SC injection, these studies found a low incidence of AEs. The MAGNOLIA and AZALEA trials demonstrate that the rates of AEs, such as increased IOP and cataract progression, were low. However, cataract progression and elevated IOP occurred and sometimes necessitated surgical or medical management, respectively. There were uniformly no reports of serious AEs, such as retinal detachment (RD) or increased IOP requiring surgery. The studies reported varied rates of AEs ranging from 18.4% to 64.3% and described different occurrence rates of IOP elevation, cataract formation, and cataract progression. This variability underscores the importance of conducting larger-scale, masked, and randomized studies with a larger number of enrolled participants. Such studies would provide more reliable and generalizable results, reduce the impact of potential confounders, and increase the overall robustness of the findings. Additionally, the efficacy and safety of SCTA cannot be compared to other therapeutics, injected either suprachoroidally or intravitreally, as all the studies to date have been single-arm interventional trials of SCTA or in comparison to sham injection.

Nevertheless, these studies collectively reinforce SCTA’s potential as an efficacious and safe therapeutic approach. As such, SCTA has been the first and only FDA-approved therapy leveraging the SCS for the treatment of ME secondary to NIU. The findings of these studies have paved the way for the study of additional therapeutics administered in the SCS, such as anti-VEGF and viral gene agents.

#### 7.1.2. Suprachoroidal Injection for Diabetic Macular Edema

DME is a common complication of diabetic retinopathy and a leading cause of vision loss. Its current first-line treatment involves IV injections of anti-VEGF agents (Ranibizumab, Aflibercept, and Bevacizumab), which have shown significant efficacy in improving vision [101]. However, these treatments have limitations, including the need for frequent injections and potential AEs related to the IV application. As an alternative, corticosteroids, such as triamcinolone acetonide (TA), have been applied as a second-line treatment due to their anti-inflammatory properties. Intravitreal triamcinolone acetonide (IVTA) has been shown to effectively reduce DME and improve vision but is associated with ocular AEs, such as increased IOP and cataract progression [45].

In this landscape, SCTA has emerged as an alternative for DME treatment. SCTA offers the potential to limit anterior exposure and possibly decrease ocular AEs. The HULK trial by Wykoff and colleagues (2018) evaluated the safety and efficacy of SCTA for the treatment of DME. They administered SCTA (4.0 mg/0.1 mL) alone or combined with IV Aflibercept (2.0 mg/0.05 mL). The combination group (10 treatment-naïve participants) received IV Aflibercept followed by SCTA, with an average of 2.6 injections. The monotherapy group (10 participants who had received previous treatment) had an average of 3.3 SCTA injections. After 6 months, the monotherapy group had a greater reduction in CMT (128 μm) compared to the combination group (91 μm), while the combination group exhibited better visual acuity gains (8.5 ETDRS letters) compared to the monotherapy group (1.1 letters). No serious or systemic ocular AEs were observed, indicating the safety of SCTA in treating DME [102]. A post-hoc analysis of the HULK trial showed that SCTA caused a measurable increase in the SCS, which returned to baseline levels 1 month following injection with no lasting anatomical impacts [4]. Similarly, the TYBEE trial by Barakat and collaborators (2021) compared SCTA with IV Aflibercept versus IV Aflibercept alone in treatment-naïve DME patients among 36 versus 35 participants, respectively. After 24 weeks, the difference in BCVA improvement was not statistically significant between groups. The combination group showed a notable advantage in that they exhibited a greater reduction in CMT and necessitated fewer prn injections compared to the monotherapy group [103].

Studies have also compared the efficacy and safety of SCTA in combination with IV Bevacizumab versus IV Bevacizumab alone. In a phase II/III randomized controlled pilot trial, Fazel and colleagues (2023) randomly assigned 66 eyes with untreated DME to receive SCTA in combination with IV Bevacizumab or IV Bevacizumab alone. They found that adding a single dose of SCTA prior to IV Bevacizumab led to significant improvements in BCVA and reductions in CMT without major ocular AEs. After 3 months, there was a significant improvement in mean BVCA, a significant reduction in mean CST, and no significant change in mean IOP, which remained at around 15 mmHg [104]. They also determined that SCTA with IV Bevacizumab was more effective in reducing CMT compared to IV Bevacizumab alone. However, BCVA changes were not directly assessed in this study, which assumed a correlation between CMT decline and BCVA improvements based on other trials [105]. After randomly assigning 136 participants to receive either SCTA or IV Bevacizumab, Anwar and colleagues (2022) determined that a single dose of SCTA resulted in greater improvements in visual acuity and a more significant reduction in CMT [106].

Comparisons have also been made between SC and IV administration of corticosteroids. In their prospective interventional study, Zakaria and collaborators (2022) randomized 45 eyes in 32 patients to receive IVTA alone or with two different doses of SCTA (4.0 mg/0.1 mL or 2.0 mg/0.1 mL). Significant improvements in visual acuity and CMT were observed in both treatment arms after 1 and 3 months. However, after 3 months, CMT started to increase, and the reduction was not significantly different compared to baseline except in the 4.0 mg SCTA group which sustained a reduction of 60.16 µm. The 4.0 mg SCTA group demonstrated the most substantial improvement in visual acuity and sustained its effect for a longer duration, thereby confirming the effectiveness of this dosage in clinical practice. The incidence of AEs, such as IOP elevation and cataract progression, did not significantly differ between the two routes. Given that CMT had nearly returned to baseline values in most patients, they recommended considering reinjection before 6 months [107].

Other research has identified that SCTA and IVTA are similarly effective at reducing CMT and BCVA at 3 months. However, IVTA has been associated with significantly higher IOP levels and a shorter duration of effect, suggesting SCTA may be a more beneficial treatment option. For instance, Shaikh and colleagues (2023) observed comparable efficacy of SCTA and IVTA in improving BCVA and CMT at 3 months. Their study included 34 patients randomly assigned to each treatment group, with a second injection administered at 6 weeks. After 1 and 6 months, both groups demonstrated statistically significant improvements in BCVA and CMT compared to baseline, but no significant differences were observed between the groups. At 3 and 6 months, there was a significant increase in IOP in the IVTA group compared to the SCTA group. Thus, both routes were equally effective, but the SC route maintained a more favorable effect on IOP. Cataract progression was also found to be slower in eyes that received SCTA [108].

SCTA has also shown promising results in DME post-vitrectomy, with improved visual acuity and reduced macular thickness. In a study by Marashi and Zaza (2022), it was observed that among 11 (1 phakic and 10 pseudophakic) eyes treated with SCTA, significant vision improvement and a 45.74% reduction in CMT from baseline was noted after 8 weeks. Importantly, no IOP elevation or cataract progression was observed [109].

Several nonrandomized, single-arm studies have also provided evidence supporting the effectiveness of SCTA in improving BCVA and reducing CST with minimal AEs [110,111,112,113,114,115]. However, a study by Tharwat and colleagues (2022) suggests that formulated posterior subtenon TA (PSTA) injection may offer better outcomes for managing rDME with reduced risk of IOP elevation. In their prospective study, 75 patients were randomly assigned to three treatment groups (SCTA (4.0 mg/0.1 mL) alone, a combination of PSTA (40 mg) formulated with VISCOAT containing sodium chondroitin sulfate (20 mg) and sodium hyaluronate (15 mg), or unformulated PSTA (40 mg)). All groups showed a significant increase in BCVA and a significant decrease in CMT at months 1, 3, and 6. However, the group receiving formulated PSTA exhibited the highest BCVA and the lowest CST 6 months post-procedure, suggesting that this may be more therapeutic due to its prolonged contact and increased diffusion through the scleral barrier [116].

There is also emerging research investigating the application of SC-administered gene therapy for DME. RGX-314 is a gene therapy product that contains an AAV8 vector encoding an antibody fragment designed to inhibit anti-VEGF. The ALTITUDE trial, an ongoing industry-sponsored, phase II, randomized, dose-escalation study, is currently exploring the efficacy, safety, and tolerability of delivering this suprachoroidally in patients with center-involved DME [117]. Approximately 100 participants will be enrolled into one of five cohorts containing different dosages of RGX-314 with or without post-procedure steroid injection. While the trial is currently recruiting participants, interim 3-month data suggest that 33% of participants in the treatment arm had a ≥2 improvement in their diabetic retinopathy severity score compared with 0% in the control arm.

Overall, the available data suggest that SCTA at a 4.0 mg/mL dose offers numerous advantages over conventional therapies for both primary and rDME. SCTA used in combination with IV anti-VEGF agents has consistently shown effectiveness in reducing macular thickness, improving visual acuity, and providing a longer duration of action compared to IV anti-VEGF treatments alone. Notably, SCTA was found to be beneficial for patients with rDME despite prior anti-VEGF injections, potentially reducing the need for multiple injections and their associated costs. TA acts as an anti-inflammatory agent, inhibiting factors, such as VEGF, that are involved in DME pathogenesis. Previous studies combining IV corticosteroids with anti-VEGF agents for rDME have demonstrated improved functional and anatomical outcomes at the expense of ocular AEs related to diffuse corticosteroid delivery [118]. In contrast, SCTA combined with anti-VEGF agents addresses both the vascular and inflammatory aspects of DME with targeted delivery and reduced anterior segment exposure. Data comparing SCTA and IVTA have shown that both routes of administration result in significant CMT and BVCA improvement. However, SCTA is associated with fewer IOP-related AEs and a longer duration of effect, indicating SCTA may be more efficacious in resolving DME through corticosteroids.

Despite promising evidence, further research is needed to explore the long-term efficacy, optimal dosing strategies, and comparative effectiveness of SCTA for the treatment of DME, including its impact on visual outcomes, durability of effect, and potential AEs. Most studies have reported significant functional and anatomical improvements in BCVA and CST after SCTA administration within a 3-month timeframe. However, ocular AEs, such as increased IOP and cataract progression, can still occur, although they may have a lower incidence compared to IV administration. Further, comparative studies examining the efficacy and safety of SCTA with other therapeutic approaches, such as PSTA, would be valuable. Additionally, larger multicenter studies with longer follow-up periods are needed to determine whether the improvements in BCVA and CMT are transient or long-lasting.

#### 7.1.3. Suprachoroidal Injection for Macular Edema Secondary to Retina Vein Occlusion

The treatment of ME resulting from RVO has been the focus of several industry-sponsored clinical trials assessing the effectiveness of combined SCTA with IV anti-VEGF agents versus IV anti-VEGF monotherapy alone. For instance, the TANZANITE trial, a phase II, multicenter, masked, industry-sponsored RCT conducted by Campochiaro and colleagues (2018), compared the combination of SCTA (4.0 mg/0.1 mL) with IV Aflibercept (2.0 mg/0.05 mL) to IV Aflibercept monotherapy. Forty-six patients were randomized to either treatment arm and received IV Aflibercept as needed over a 3 month study period. Results showed that the combination therapy significantly reduced the need for retreatment, with 23 retreatments in the combination group versus 9 in the monotherapy group. Moreover, a higher percentage of patients did not require retreatments (78% versus 30%, respectively). The combination group also led to greater visual acuity improvement (18.9 versus 11.3 ETDRS letters in month 3) and a decrease in CST from baseline (731.1 μm to 284.7 μm at month one, stable at 2 and 3 months). In contrast, the IV Aflibercept group had an increase in CST at 2 and 3 months. Additionally, the combination group exhibited a higher percentage of edema resolution (78.3% versus 47.8%) at month 3. Although four patients in the combination group experienced elevated IOP, that was resolved with topical anti-glaucoma agents [119]. Extension data from the study indicated that 74% of patients in the combination group did not require retreatments over a 9-month period compared to 17% in the control arm [120].

Another noteworthy study, the phase III SAPPHIRE study, also compared the combination of SCTA (4.0 mg/0.1 mL) with IV Aflibercept therapy (2 mg/0.05 mL) to IV Aflibercept monotherapy in 460 eyes with RVO. After 8 weeks, approximately 50.0% of patients in both groups reported a significant improvement of ≥15 ETDRS letters in BCVA. However, no other benefits were observed in the combination arm, leading to the study’s discontinuation. Nevertheless, preliminary data from 128 patients in the combination group and 127 patients in the control group revealed that the combination procedure had a favourable safety profile, as only one case of RD and one case of vitreous hemorrhage were reported. These findings indicate that SCTA and IV Aflibercept combination therapy was well-tolerated, without significant ocular AEs [121].

A phase III, randomized, masked RCT (TOPAZ) was designed to investigate if SCTA in combination with IV Ranibizumab or IV Bevacizumab was superior to IV Ranibizumab or IV Bevacizumab alone. Treatment groups received either a combination therapy of IV Ranibizumab (0.5 mg/0.05 mL) with SCTA (4.0 mg/0.10 mL) or IV Bevacizumab (1.25 mg/0.05 mL) with SCTA (4.0 mg/0.10 mL). The control arms received either IV Ranibizumab or IV Bevacizumab, followed by a sham SC procedure. However, the trial was prematurely stopped due to the results of the SAPPHIRE trial findings [122].

In addition to industry-sponsored trials, independent studies have also examined the effectiveness and safety of combining SCTA administration with IV anti-VEGF agents. Nawar (2022) conducted a prospective randomized study on 60 patients with branch retinal vein occlusion (BRVO) to explore this treatment approach. The patients were divided into two groups, one receiving combined IV Ranibizumab with SCTA and the other receiving IV Ranibizumab alone. Both groups received monthly Ranibizumab injections as needed during the 12-month study period. Participants in the combination arm required fewer injections (2.47 ± 1.2) compared to those in the monotherapy arm (4.4 ± 1.5). At 12 months, both groups demonstrated significant reductions in CMT, along with significant improvements in BVCA. The combination group showed more significant BVCA improvement at 6 and 12 months [123].

Studies investigating SCTA as a monotherapy for RVO-associated ME have also shown promising results in terms of BCVA improvement and CST reduction. Recent research conducted by Ali and colleagues (2023) investigated SCTA as a standalone treatment in 16 patients with ME secondary to RVO. Their findings demonstrated that 68.7% of patients had a BCVA improvement of ≥15 letters by week 1 and a range of 50% to 62.5% showing this improvement during months 1 to 3. There was also a notable CST reduction throughout the follow-up period. One patient experienced elevated IOP of ≥20 mmHg in the first month, but their IOP decreased by the second month [124]. Similarly, Muslim and colleagues (2022) studied the application of SCTA (4.0 mg/0.1 mL) in 45 patients with unilateral RVO-associated ME. A statistically significant improvement in BCVA was observed after 1 month and with further improvement at 3 months. There was also a significant reduction in central retinal thickness (CRT) after 3 months [125].

Stanislao and colleagues (2012) conducted a prospective study of six eyes of six patients with central retinal vein occlusion (CRVO), BRVO, or diffuse DME accompanied by severe refractory subfoveal hard exudates (SHE). Participants received a single SC infusion of Bevacizumab and TA administered into the submacular SCS using a microcatheter at the pars plana. Four eyes showed an improvement of ≥2 lines in BCVA, while two eyes remained stable. By 1 to 2 months, SHE was almost completely resolved in all eyes, and ME was significantly reduced with no surgical or post-injection complications reported [126].

Based on these findings, the benefits of a combination of SCTA and IV anti-VEGF therapy for RVO-associated ME include fewer retreatment needs, improved visual acuity, anatomical improvements, and resolution of subretinal exudates. Corticosteroids have shown efficacy in addressing inflammation associated with RVO by targeting molecules that affect vascular permeability and inflammation. The SCS offers a targeted pathway for drug delivery to manage ME secondary to RVO. However, given the lack of large-scale, independent, and multicenter studies examining the application of SCTA in the context of ME secondary to RVO, additional research is required to confirm the optimal combination therapies, most effective drug combinations, and long-term efficacy and safety of SCTA. To address this knowledge gap, an ongoing clinical trial is taking place in Egypt that is investigating the long-term effects of SC injection, including ocular hypertension, cataract progression, and ME resolution, in the treatment of RVO alongside other retinal diseases such as Vogt Koyanagi Harada disease and DME. The trial is still in the recruitment phase, with results pending study completion [127].

#### 7.1.4. Suprachoroidal Injection for Post-Operative/Pseudophakic Cystoid Macular Edema

SCTA has also shown promise as a potential treatment for pseudophakic cystoid macular edema (PCME), a common post-operative complication of cataract surgery. A study by Tabl and colleagues (2022) demonstrated the efficacy of SCTA and IVTA in reducing CFT and improving visual acuity in pseudophakic patients with rDME caused by the epiretinal membrane. They injected SCTA (4.0 mg/0.1 mL, in 13 eyes) or IVTA (4.0 mg/0.1 mL, in 10 eyes) with results consistent with Zakaria and collaborator’s findings on the significant improvements in CFT (see Figure 5) and BCVA with a 4.0 mg dose of SCTA. The IVTA group had significantly higher elevations in IOP in the first month (15 mmHg) compared to the SCTA group (12 mmHg). Furthermore, by the third month, the IVTA group still exhibited significantly higher IOP levels (18 mmHg) compared to the SCTA group (14 mmHg), indicating a sustained difference between the two groups [128]. Similarly, Zhang and colleagues (2022) injected SCTA (0.2 mL of 40 mg/mL) in 20 eyes of 20 patients with CME and PCME, resulting from various conditions such as BRVO, CRVO, DME and previous epiretinal membrane (ERM) peeling surgery. Optical coherence tomography (OCT) examination confirmed drug delivery as determined by SCS expansion near the injection site. The injections led to significant improvements in BCVA and CST without significant differences in IOP. No complications, such as cataract induction, hemorrhage, retinal detachment, or endophthalmitis, were observed during the 3-month study period. The authors proposed that the anterior SCS is the most accessible location for injection, in alignment with previous animal and human studies. However, the long-term efficacy and safety of SCTA for CME could not be established due to the lack of participant follow-up [129].

Other case studies by Oli and Waikar (2021) and Marashi and Zazo (2022) also reported positive outcomes with SCTA for PCME, including improved BCVA and decreased CMT [130,131]. Additionally, a clinical trial is currently investigating the impact of SCTA on CME caused by Irving-Gass syndrome following cataract surgery [132].

Overall, preliminary research suggests that SCTA could be effective for managing PCME. The results align with previous investigations that treated ME secondary to uveitis, DME, or RVO with SCTA using a microinjector approach. However, more extensive randomized studies are needed to evaluate the long-term efficacy and safety of SCTA for post-operative complications such as PCME.

### 7.2. Photoreceptor Loss

#### 7.2.1. Suprachoroidal Injection of AAV Vectors for the Treatment of Inherited and Acquired Retinal Disorders

Preclinical studies have shown that SC delivery of adeno-associated vectors (AAV) is a promising technique for treating inherited and acquired retinal diseases. Peden and collaborators (2011) conducted a pioneering study using a microcatheter to deliver AAV5 (100 µL of sc-AAV5-smCBA-hGFP vector at a concentration of 4.5 × 10^13^ vector genomes/mL) into the SCS of eight healthy rabbits. The treatment was well tolerated, with no reports of serious AEs. Analysis of whole-mounted treated eyes 6 weeks post-injection revealed robust transfection, evidenced by the presence of GFP expression in the choroid, the RPE, photoreceptors, and retinal ganglion cells. In contrast, the control group did not exhibit any GFP expression. The authors concluded that the microcatheter approach for SC AAV delivery demonstrated safety, tolerability, and effective gene transfer to target areas [65]. Similarly, Martorana and colleagues (2012) compared the gene transfer of AAV2, AAV5, and AAV2, containing three tyrosine-phenylalanine mutations on the capsid surface [AAV2(triple)]. The efficiency of SC and subretinal transduction was further compared in rabbits. Immunostaining showed that GFP expression was observed in all eyes that received vitrectomy/subretinal or SC injections, with AAV2 producing the strongest GFP expression. There was intermediate expression with AAV2 treatment and minimal expression with AAV5 treatment, unlike Peden’s findings. Transduction profiles were not affected significantly by the vector concentration [133]. Both studies demonstrated the feasibility of delivering AAV vectors through SC injection, although outcomes varied depending on the serotypes used. Importantly, this approach reduced the surgical risks associated with the current approach of conventional 3-port PPV followed by subretinal treatment.

Woodard and colleagues (2016) compared different routes of AAV2 administration in mice, including intrastromal, intracameral, IV, subretinal, and SC injections. In their mouse model, AAV2 was used to deliver a genetic construct containing a promoter region derived from cytomegalovirus (CMV) alongside a GFP reporter gene. Examination with fundoscopy and OCT assessed the anatomical impact of the injections at the time of administration, and transduction was evaluated after 6 weeks using fundoscopy and histological analysis of whole globes. Transduction was observed in multiple ocular structures, including the stroma, ciliary body, retinal ganglion cells, outer retina, and the RPE, irrespective of delivery route. Notably, SC injections demonstrated transduction across multiple retinal layers throughout the entire retina. This ability to transduce retinal layers without inducing a temporary RD led the authors to conclude that SC delivery may offer unique advantages over subretinal delivery [134].

Recent studies have investigated the effectiveness of SC AAV delivery in animal models using a conventional hypodermic needle and free-hand method. Ding and colleagues (2020) used this method to inject a GFP-reporter gene with RGX-314, an AAV8 vector expressing a VEGF-neutralizing protein, into the SCS in animal models. India ink injection into the SCS confirmed its spread throughout the posterior segment without entering the RPE or retina. Two weeks later, treated eyes displayed robust fluorescence in the RPE and outer retina on the injected side, extending to the opposite side of the eye. Immunohistochemical staining confirmed GFP presence in the RPE, photoreceptor cell bodies, inner segments, and outer segments, with stronger staining near the injection site. Conversely, subretinal injection resulted in strong fluorescence and GFP staining only on the injected side, with minimal staining in remote quadrants. Rats that received a second SC injection of AAV8 showed increased GFP expression compared to a single injection. The study also compared SC delivery of AAV8, AAV9, and AAV2 serotypes, with AAV8 and AAV9 displaying strong GFP expression in the injected eye quadrant, while AAV2 exhibited limited fluorescence in the far periphery. Serum albumin, an endogenous marker for vascular leakage, was used to assess retinal vascular permeability. Eyes treated with SC or subretinal delivery of RGX-314 showed significantly lower vitreous albumin levels compared to control eyes injected with AAV8, indicating the suppression of VEGF-induced vasodilation and vascular permeability by RGX-314. Additionally, the study confirmed similar levels of anti-VEGF Fab protein in the retina, the RPE, and the choroid between SC and subretinal routes [135].

Ding and colleagues (2020) also conducted a study comparing SC delivery of AAV2tYF-CBA-hGFP, AAV2tYF-GRK1-hGFP, AAV5-GRK1-hGFP, or AAV2-CBA-hGFP in 65 Norway brown rats. Peak GFP expression was achieved by each vector at 2 weeks, with AAV2tYF showing further increase between weeks 2 and 4. AAV2tYF exhibited stronger and more widespread GFP expression, extending approximately ¼ of the circumference of the eye in the RPE and all layers of the retina. Significant transduction of photoreceptors and inner retinal cells was also observed with AAV2tYF-GRK1-GFP and AAV5tYF-GRK1-GFP via the SC route. AAV2tYF-CBA provided significantly greater transduction than AAV2-CBA after SC injection. While not as extensive as AAV8 and AAV9, AAV2tYF-CBA resulted in more transduction of inner retinal cells. AAV2tYF-GRK1 demonstrated superior and more extensive transduction of photoreceptors compared to AAV5-GRK1. These findings support the potential of SC injection of AAV2tYF-CBA and AAV2tYF-GRK1 for efficient transduction of retinal cells, particularly photoreceptors [136].

Yiu and colleagues (2020) conducted a study on non-human primates to compare the efficacy of SC, subretinal, and IV gene delivery using AAV8 carrying an enhanced GFP sequence. SC injection resulted in widespread transgene expression in the RPE, while subretinal delivery showed focal transduction in the RPE, photoreceptors, and some ganglion cells near the injection site. IV injection led to scant peripapillary GFP expression in cells, potentially astrocytes or Müller glia. Other studies comparing SC delivery of AAV serotypes with other routes of transmission in animal models have confirmed that SC administration may be preferable due to their widespread transduction and lack of associated retinal complications. However, Han and colleagues (2020) and Tian and colleagues (2021) investigated the use of different AAV serotypes for gene transfer in animal models, which showed that transduction, estimated by GFP expression, varied among serotypes [59,137].

Further, the initial enthusiasm for gene therapy may be tempered by emerging evidence of AAV-associated inflammation. For instance, Yiu and colleagues discovered that SC AAV8 delivery resulted in transient expression, peaking at month 1 with a subsequent decline by months 2 and 3. This decline was attributed to cellular damage and the phagocytic activity of local inflammatory cells. In contrast, subretinal and IV delivery showed lower localized chorioretinitis, although IV administration produced a stronger systemic humoral immune response [24]. A subsequent study by Ching and collaborators (2021) demonstrated that SC delivery induced a lower systemic immune response compared to IV delivery but higher elevations in IOP compared to subretinal delivery. These results were anticipated due to the SCS being located outside the blood–retinal barrier, rendering it susceptible to immune surveillance cells. While the study refrained from extensive immunosuppression to assess the natural immune response to SC AAV8 delivery, future research could explore the influence of corticosteroids. The authors also observed reduced transgene expression after 2 and 3 months, likely due to phagocytic activity of infiltrated macrophages and leukocytes observed at 1 month [138]. In a separate study, Wiley and collaborators (2023) examined the extent and retinal pattern of AAV-associated inflammation in rats following the administration of five distinct AAV vectors (AAV1, AAV2, AAV6, AAV8, and AAV9). AAV2 and AAV6 consistently induced higher levels of inflammation levels compared to control groups across all delivery routes. Specifically, AAV6 triggered the highest inflammation when delivered using the SC route. AAV1 exhibited significant inflammation with SC delivery but minimal inflammation with IV delivery. AAV1, AAV2, and AAV6 also activated adaptive immune cells. AAV8 and AAV9 caused the least inflammation regardless of the delivery route. Interestingly, the amount of inflammation was not correlated with transduction and GFP expression [139].

Over the past decade, gene therapy using AAV vectors has shown promising results in animal trials, but delivery methods need further investigation to reduce risks and for optimal targeting. By enabling the transduction of multiple retinal layers without the risk of complications such as RD, this route may enhance the efficacy and distribution of gene therapy in the retina. However, translating these findings into clinical practice presents several challenges. First, the precise targeting of specific retinal regions using the SC approach remains unclear. These studies demonstrate that SC injections can treat larger peripheral areas affected by retinal diseases. However, some studies have indicated that SC delivery may result in less exposure to the inner retinal layer, minimizing transduction of retinal ganglion cells in comparison to IV administration. Thus, SC gene delivery using a microneedle may lack regional specificity and require optimization for macular targeting. Strategies such as using a “pushing” formulation that exerts pressure to facilitate movement of therapeutics, iontophoresis, collagenase to expand the SCS, higher injection volume, or catheter-based delivery could improve SC targeting [140]. Another challenge is the potential for immune responses and inflammation associated with AAV gene therapy. Although research suggests that SC AAV injections are associated with reduced systemic inflammation, they can potentially induce local inflammation. For instance, Yiu and colleagues found that persistent transgene expression in scleral cells following SC AAV administration may decrease over time due to the presence of inflammatory cells, leading to disruption of the RPE and photoreceptor segments [24]. Interestingly, IV AAV injections resulted in a stronger systemic immune response compared to subretinal or SC gene delivery, highlighting the different immune consequences of AAV exposure in different compartments surrounding the outer blood–retinal barrier. Further research is needed to explore local inflammatory responses associated with SC gene administration. Additionally, advances in AAV technology, such as the application of multiple AAV vectors simultaneously and intein-mediated protein trans-splicing, should be evaluated for SC delivery.

#### 7.2.2. Suprachoroidal Injection of DNPs for the Treatment of Inherited and Acquired Retinal Disorders

SC injection of nanoparticles is an emerging approach for ocular gene therapy. DNA nanoparticles (DNPs), composed of DNA molecules, can be used to deliver therapeutic genes or gene-editing tools into target cells [59]. Researchers commonly use DNPs carrying a luciferase gene to measure luciferase activity and assess gene delivery efficiency. Kansara and colleagues (2019) performed SC injection of ellipsoid-shaped DNPs, rod-shaped DNPs, or saline in non-human primates, alongside SC injection of analogous DNPs and subretinal injection of rod-shaped DNPs in rabbits. Luciferase activity was observed in the retina, choroid, and the RPE. Ellipsoid-shaped DNPs showed persistent luciferase activity up to day 22, while rod-shaped DNPs declined in non-human primates. In rabbits, both SC-injected rod and ellipsoid-shaped DNPs showed similar luciferase activity after 1 week. The study demonstrated successful transfection of chorioretinal cells using SC-delivered DNPs [137].

In a follow-up study, Kansara and collaborators (2020) compared SC and subretinal injections of DNPs in rabbits. Microneedle-based SC administration of DNPs was also well-tolerated and effective in transfected chorioretinal tissues. SC injection provided greater surface area coverage and aided in the transfection of the peripheral retina. DNPs injected into the SCS showed minimal intraocular toxicity, while subretinal injections displayed ocular toxicity. The study established the potential of nonviral-based gene delivery to the chorioretina via the SC administration [139].

SC delivery of poly (β-amino ester)s nanoparticles (PBAE NPs), a biodegradable polymer used for gene delivery, has also been explored. In a study involving Brown Norway rats, performed by Shen and colleagues, SC injections of PBAE NPs containing various plasmids showed widespread GFP expression throughout the retina. However, this was less intense in the RPE and photoreceptors compared to AAV8 injections. Widespread lateral and radial penetration of polymeric NPs via SC delivery was attributed to a transient pressure increase induced by the injected volume into the SCS space. However, SC injection of PBAE NPs containing a VEGF expression plasmid caused severe subretinal neovascularization, similar to AMD. Conversely, SC injection of PBAE NPs containing a VEGF-binding protein suppressed VEGF-induced retinal vascular leakage and neovascularization, demonstrating therapeutic potential. Expression was quite strong 2 weeks after injection and was maintained for at least 8 months. Compared to a single SC injection of NPs containing pEGFP-N1, three injections resulted in a five-fold increase in ocular expression of GFP, demonstrating the feasibility of increasing expression using repeated injections [140].

Overall, SC injection of nanoparticles holds promise for treating various retinal diseases. Unlike AAVs, nanoparticles offer a nonviral-based gene therapy option that can be repeated over time, allowing for multiple treatments if needed. They can also transfer large genes common in inherited retinal disorders, such as Stargart’s macular dystrophy (SMD) [59]. However, they may result in variable gene expression intensity and neovascularization risk. AAVs, on the other hand, may trigger elevated immune responses due to pre-existing antibodies against AAV capsid antigens. As research progresses, SC injection of nanoparticles may become a valuable therapeutic strategy to address the underlying genetic causes of retinal diseases.

#### 7.2.3. Suprachoroidal Injection for the Treatment Dry-Aged Macular Degeneration and Stargardt’s Macular Dystrophy

Ongoing research is exploring the use of non-retinal-derived stem cells, specifically mesenchymal stem cells, for the treatment of degenerative retinal diseases. These stem cells secrete various factors that have anti-apoptotic, anti-inflammatory, immunomodulatory, and angiogenic effects, providing trophic support for damaged retinal cells [141]. Recent studies have explored SC delivery of mesenchymal stem cells for dry AMD and SMD. In a prospective study conducted by Kahraman and colleagues (2021), eight patients with advanced-stage dry-type AMD and SMD underwent SC implantation of adipose tissue-derived mesenchymal stem cells (ADMSCs) in their worst eye. All patients experienced improvements in visual acuity, visual field, and multifocal electroretinography (mfERG) with no serious complications. These improvements persisted throughout the 1-year follow-up period, accompanied by choroidal thickening, indicating increased choroidal perfusion. The proximity of SC implantation allows the produced growth factors to enter into the choroidal flow, enhancing interactions with retinal cells. However, the study included only patients with severe visual loss [142]. Thus, future research should focus on larger patient cohorts at earlier stages of the disease to refine treatment timing, graft replacement strategies, and delivery methods. Nevertheless, these findings offer promising evidence for effective treatment of degenerative retinal diseases.

#### 7.2.4. Suprachoroidal Injection for the Treatment of Retinitis Pigmentosa

Retinitis pigmentosa (RP), a collection of inherited retinal disorders, is characterized by progressive photoreceptor loss, leading to significant visual impairment. A wide spectrum of genetic mutations challenge the development of efficacious treatments for RP. Expanding our understanding of potential therapeutic strategies, as demonstrated in Figure 6, remains a critical objective. We have previously highlighted the potential of SC injection as a delivery mechanism for gene therapy in the management of RP. The focus of the current section is to extend this discussion to illustrate the application of SC injections in cell therapy, specifically the use of umbilical cord-derived mesenchymal stem cells (UCMSCs).

Oner and colleagues conducted two studies assessing the effects of UCMSC implantation on RP patients. In their first study, significant improvements were observed in mean BCVA and visual field scores over a 12-month period. Notably, the treatment also led to an improvement in disease score and grade [143]. In their second study, which focused on pediatric RP patients, UCMSC implantation resulted in significant enhancements in BCVA, visual field examination, and mfERG measurements in all 46 eyes of patients. No systemic or ocular complications were reported [144]. However, the use of SC mesenchymal spheroidal stem cell implantation is still being evaluated, and study results are pending [145]. Further research is required to gain a comprehensive understanding of the potential advantages and limitations of this treatment approach for RP.

#### 7.2.5. Suprachoroidal Injection for Solar Retinopathy

Marashi et al. (2021) published a case report describing a 17-year-old female with a sudden scotoma due to solar retinopathy. The patient received a single SCTA (4.0 mg/0.1 mL) injection with a custom-made needle. After 1 week, the patient’s BCVA improved from 0.1 to 1.0, and her scotoma disappeared. A mild elevation in IOP to 28 mmHg was observed at 7 weeks, which resolved to normal limits with topical anti-glaucoma agents. After 4 weeks, there was a full recovery in her BCVA, and OCT demonstrated anatomical improvement in the ellipsoid zone layer. No serious AEs were reported [146]. Overall, the implications of this case suggest SCTA may be a promising therapeutic option for solar retinopathy, leading to significant improvements in visual acuity and anatomical changes without serious or unmanageable AEs. However, further research is needed to establish the efficacy, safety, and long-term outcomes of this approach in larger patient cohorts.

### 7.3. Choroidal Neovascularization

#### 7.3.1. Suprachoroidal Injection for Solar Retinopathy

In individuals with choroidal neovascularization (CNV), the production of VEGF triggers abnormal and chronic angiogenesis. Consequently, sustained suppression of VEGF is required to effectively manage CNV. For this purpose, VEGF inhibitors such as Ranibizumab, Aflibercept, and Bevacizumab are typically intravitreally injected for CNV treatment. However, the need for frequent injections, as often as monthly, can be burdensome for patients and impose substantial costs on the healthcare system [147,148]. Additionally, IV injections of anti-VEGF agents are associated with AEs such as endophthalmitis, cataract, or RD [149,150]. This situation has driven research toward newer delivery methods and longer-lasting alternative medications.

To optimize drug delivery to the macula, Tran and colleagues conducted a preclinical study in 2017. A total of 39 surgical pig models with surgically induced CNV were injected with either 2.5 mg IV Bevacizumab, 1 mg IV Pazopanib, 300 μg IV hI-con1, or 1 mg SC Pazopanib, with comparable SC and IV vehicle controls. This study used novel anti-VEGF agents, such as Pazopanib and ghI-con1, to evaluate their efficacy. IV Pazopanib resulted in smaller mean height measurements of CNV type 2 lesions compared to the SC Pazopanib, and these measurements were statistically smaller than controls. For eyes treated with IV Bevacizumab, there was only a small decrease in the height of the lesions in comparison to controls. There were no significant differences between the surface area of CNV lesions between the three treatment groups. While IV-injected hI-con1 resulted in lesions that were thinner than controls, these results were not statistically significant. Their study concluded that IV Pazopanib, and, to a lesser extent, hI-con1, inhibits induced CNV lesions in pig models [151]. Given the similar properties between Pazopanib and TA, Tran and colleagues hypothesized that this medication would be a well-suited SC injection. Surprisingly, they found that IV injections yielded more significant inhibition of CNV lesions than SC injection. This could be due to Pazopanib’s limited solubility, potentially resulting in adequate distribution to the posterior segment. Additionally, the low solubility might lead to a slower drug distribution, causing an insufficient amount to reach the posterior eye. Additionally, there could have been underdosing if some material remained in the syringe, in addition to dosing variations per injection. Consequently, comprehensive studies measuring the precise amount of Pazopanib injected into the SCS and analyzing its pharmacokinetics and distribution in animal models are necessary before advancing to human trials [151].

On the other hand, Mansoor and collaborators (2012) concluded that Bevacizumab (Avastin, 1250 μg/50 μL) injected into the SCS reached excellent levels in the choroid, sclera, and retina but exhibited a rapid decline in the choroid after only 1 day. They attributed this outcome to the suboptimal formulation of SC Bevacizumab, which failed to effectively target the posterior eye segments in a sustained-release matter [152]. Earlier studies of porcine models also demonstrated the rapid clearance of Bevacizumab when injected suprachoroidally in comparison to intravitreally [153]. These findings emphasize the need for optimizing drug formulations for SC injection, potentially through methods such as increasing injectate viscosity and particle size or using novel vehicles to create sustained-release formulations. Interestingly, Tyagi and colleagues (2013) successfully formulated a gel network using light-activated polycaprolactone dimethacrylate and hydroxyethyl methacrylate. This network enabled the sustained release of Bevacizumab for over 4 months when injected into the SCS in animal models. This sustained release approach did not compromise the mechanism of action of Bevacizumab [66]. Similarly, Jung and colleagues (2022) demonstrated that an in-situ forming hydrogel comprised Bevacizumab and HA crosslinked within 1 h of injection into the SCS of a rabbit allowed for slower release as the hydrogel underwent biodegradation. The degradation happened for over 6 months, and ophthalmological examination, fundoscopy, imaging, histological analysis, and IOP assessments confirmed it was well-tolerated [80].

Acriflavine is recognized for its ability to suppress neovascularization by reducing the transcriptional activities of hypoxia-inducible factor-1 and factor-2 involved in pathogenesis. Zeng and colleagues conducted a study using a laser-induced rat model of CNV in 2017 and found that 300 ng of SC Acriflavine spread throughout the retina and choroid by day 1 and was maintained for 5 days. Additionally, this treatment caused a CNV reduction 14 days after Bruch’s membrane rupture. They also examined intraocular injection of 100 ng and extraocular 0.5% drops, but there was no formal comparison between different routes of administration. They concluded that various modes of Acriflavine delivery have the potential to be used for CNV treatment pending further research [154]. Building on this, Hackett and collaborators (2020) developed a sustained delivery method that increased the delivery time of Acriflavine into the SCS using poly (lactic-co-glycolic acid) microparticles for up to 60 days. They found IV and SC injections of Acriflavine using this microparticle suppressed CNV for 9 weeks in mice and 18 weeks in rats, respectively. Notably, IV injection of 38 μg Acriflavine resulted in a modest reduction in full field electroretinogram function, while SC injection resulted in no electroretinogram functional, IOP, or retinal changes over 28 days [84].

Emerging methods, such as SC electrotransfer, present alternatives for drug delivery into the SCS. Touchard and colleagues (2012) found that SC electrotransfer of a VEGFR-1 (sFlt-1)-encoding plasmid significantly inhibited laser-induced CNV in rats at 15 days. No retinal or vascular AEs were observed, suggesting that this minimally invasive method opens the door for novel research in the retinal disease treatment [88].

Animal studies have demonstrated the efficacy of established and novel anti-VEGF agents in the SCS for the treatment of CNV, including Bevacizumab, Pazopanib, and Acriflavine. Notably, sustained-release formulations have prolonged the efficacy of SC Bevacizumab and Acriflavine without compromising their safety. However, human trials are essential to confirm their safety and effectiveness, with further comparative studies needed to assess the suitability of various drugs. It is also crucial to recognize that animal studies have limitations, particularly their inability to replicate aging-related CNV lesions driven by VEGF, which are typically type 1 lesions seen in humans, unlike the type 2 lesions induced by Tran and colleagues [151]. Resultantly, clinical studies are imperative to bridge this gap [80]. Nonetheless, the relevance of porcine models should not be underestimated due to their anatomic similarities to humans, including comparable scleral thickness size, ocular blood flow, and RPE characteristics [155].

#### 7.3.2. Suprachoroidal Injection for Choroidal Neovascularization Secondary to Neovascular Age-Related Macular Degeneration

Gene therapy is a promising treatment for inherited and acquired retinal diseases, with its use being explored for CNV. In a 2022 phase II clinical trial of 50 patients, Khanani used an AAV8 vector to deliver anti-VEGF fab transgene with the goal of creating continuous therapy in the eye. Patients were randomized to receive SC RGX-314 at levels of 2.5 × 10^11^ and 5 × 10^11^ genomic copies/eye or monthly 0.5 mg IV Ranibizumab. Patients were found to have stable BCVA and CRT at 6 months, with a meaningful reduction in anti-VEGF treatment burden (>70%). In both groups, 29% and 40%, respectively, received no anti-VEGF injections over 6 months following RGX-314 administration. Treatment-related AEs were mild, with 23% of participants experiencing mild intraocular inflammation at similar rates for both dose levels that resolved with topical corticosteroids. While the full results of this study remain unpublished, this approach could transform the landscape of nAMD treatment by offering an alternative regimen with reduced injection frequency [156,157].

Another area of investigation centers on the safety of SC Bevacizumab and TA for resolving treatment-resistant nAMD. Using a new microcatheter, Tetz and collaborators injected a combination of Bevacizumab and TA in the SCS in 21 eyes. After 6 months, they observed no serious intraoperative or postoperative complications. IOP elevation was experienced by 4.76% of participants at 3 months that normalized with medical treatment, and an increase in nuclear sclerotic cataracts was noted in 10.5% [158]. Similarly, a phase I clinical trial by Morales-Canton and colleagues (2013) injected four patients with CNV secondary to wet AMD with 100 µL of Bevacizumab. While patients reported moderate pain, there were no serious AEs, IOP elevation, nor need for rescue therapy at 2 months [159].

Patel and colleagues compared the effects of SC saline to 40 mg/mL SC Aflibercept in a laser-induced CNV rat model. The study revealed a notable and significant reduction in the neovascular leak area on fluorescein angiography at 21 days in those treated with SC Aflibercept [160]. Another molecule, CLS011A, has anti-VEGFR and anti-PDGFR binding properties, making it a promising new candidate for CNV treatment [92]. Kissner and colleagues (2016) injected 4 mg SC CLS011A into the eyes of rabbits and found this to be well tolerated until day 91. Over 60% of the molecule remained in the sclera, choroid, and RPE at this time point. There were no signs of toxicity and no detectable drug levels in the plasma or aqueous humor. The drug was present for the full study duration in the following areas in order from the highest concentration to the lowest: sclera; choroid; RPE; retina; and vitreous humor [161].

The safety of Axitinib, a protein kinase inhibitor that also acts as an anti-VEGF agent, is currently being assessed in a multi-center study for the treatment of nAMD at doses of 0.03, 0.10, 0.50, and 1.0 mg injected into the SCS following IV 2 mg Aflibercept in 27 eyes for 12 weeks. Preliminary safety data show that all doses were well-tolerated with no treatment-related serious AEs. Final safety data are anticipated to be released later in 2023 [162]. Further, an ongoing extension study is underway, which aims to evaluate long-term outcomes for an additional 12 weeks [163]. Before these human clinical trials, Axitinib’s safety and drug characteristics were tested in laser-induced CNV animal models by two separate studies. Both studies revealed favorable tolerance and no detectable presence of the drug in plasma or aqueous humor. Moreover, sustained high levels of Axitinib were observed in the sclera, choroid, RPE, and vitreous humor for an extended period. In a rat CNV model, 40% of eyes showed improvement by day 21 in contrast to the saline-injected group. Meanwhile, in the pig CNV model, a statistically significant reduction in fluorescein leakage was observed at weeks 1 and 2 when compared to the saline-injected group [90,92].

Given the successful outcomes of SC therapeutic agents in treating ME secondary to NIU, DME, and CNV, it is unsurprising that well-established corticosteroids and anti-VEGF agents also offer promise for SC treatment of nAMD with minimal AEs. To advance our understanding, larger and longer multicenter trials are needed to assess the safety and feasibility of SC Bevacizumab and TA. While animal studies have demonstrated Aflibercept’s efficacy and CLS011A’s favorable pharmacokinetic profile, clinical trials are required to confirm their safety and effectiveness. As discussed, a multi-center phase I/II study for the treatment of nAMD with SC Axitinib was initiated based on encouraging data from animal studies. While the preliminary safety results of this clinical trial are promising, the final efficacy and safety results, along with the extension safety study data, are critical to determine if SC Axitinib is a viable treatment option for nAMD in the long term [162,163]. In addition, novel research assessing the anatomical and functional effects of SC RGX-314 for nAMD highlights exciting new advancements that may reduce the need for frequent injections [156,157]. After the results of these clinical trials are released, comparative studies should be leveraged to identify the safest and most efficacious pharmacological agent for long-term nAMD treatment. Emerging research is also exploring novel therapeutic agents. For instance, an ongoing phase I clinical trial aims to assess the use of an integration-deficient lentiviral vector, BD311, to deliver a VEGF antibody gene for the treatment of ocular diseases characterized by CNV, such as nAMD, DME, and ME following RVO. The goal of this study is to achieve constant anti-VEGF activity by delivering the gene to the posterior segment of the eye, suppressing CNV [164].

The separation between the RPE and the innermost part of Bruch’s membrane is known as retinal pigment epithelial detachment. Many chorioretinal diseases, such as nAMD, can lead to this pathology alongside idiopathic causes. Recently, Datta and colleagues studied the efficacy and safety of two 0.1 mL SC injections of Bevacizumab administered 1 month apart in 30 patients with serous pigment epithelial detachment for 8 weeks. BCVA improved for all patients 1 week after the first injection, which was maintained and statistically significant at 8 weeks post-injection. There was an objective decrease in pigment epithelium detachment at 2 weeks post-injection, with a statistically significant decrease in mean height of the pigment epithelium detachment at 6 weeks. IOP rose transiently after injection, and patients were treated with 500 mg oral acetazolamide. Patients noted more pain in comparison to IV injections, but no other AEs were reported [165]. Considering the absence of established treatment guidelines for serous pigment epithelial detachment, as well as its limited response to existing treatment options, the findings of this study offer promising prospects for the management of this condition pending larger studies of longer duration.

### 7.4. Suprachoroidal Injection for Retinal Detachment

The safety and efficacy of SCTA have been well-established in preclinical and clinical studies for ocular diseases, such as ME, secondary to NIU and DME, leading to further investigation of its application for RD. Traditionally, surgical intervention has been the cornerstone for managing rhegmatogenous RD, but addressing the underlying inflammatory process associated with this approach has been deemed beneficial. Systemic steroid therapy may not be universally applicable due to patients’ medical comorbidities, such as uncontrolled diabetes and hypertension. Topical steroids pose an increased risk of globe perforation and result in inconsistent drug bioavailability. Likewise, IV corticosteroids can raise IOP and increase the risk of cataract formation and progression [166].

Given these challenges, the utilization of SCTA injections has emerged as a potential alternative to address the limitations associated with other treatment options. Tabl and collaborators (2022) conducted the first clinical trial assessing the use of SC injection for the treatment of RD via injection of SCTA. The study encompassed six eyes with serous retinal detachment caused by Vogt–Koyanagi disease, with untreated eyes serving as controls. Notably, all patients were in the acute phase of the disease and concurrently receiving systemic steroids. The trial demonstrated significant improvements in BCVA and central foveal thickness (CFT) in eyes treated with SCTA at both 1 and 3 months, with no significant difference in IOP between the treated and untreated eyes. These findings confirm the potential of SCTA as an effective adjunctive treatment, along with oral steroids, for managing serous RD due to Vogt–Koyanagi disease [167]. Similarly, another study by Kohli and collaborators (2022) showcased the success of using 4.0 mg SCTA prior to vitrectomy and scleral buckle surgery in 10 patients with serous choroidal detachment associated with rhegmatogenous RD. This prospective, non-comparative study revealed that SCTA resulted in 50% of eyes having >50% reduction in fluid by day 3 and 20% by day 5. In total, only 30% of eyes required surgical drainage before proceeding with vitrectomy. While one eye (10%) experienced a transient increase in IOP to 30 mmHg, which was managed with topical anti-glaucoma medications, no other treatment-related AEs were reported [166].

While preliminary studies show that SCTA may be a promising adjuvant treatment for two types of RD, larger comparative studies are needed to assess if the long-term outcomes of this additional procedure are worth the extra cost and time required to perform it. Currently, clinical research on SCTA is limited to serous RD due to Vogt–Koyanagi disease and serous choroidal detachment associated with rhegmatogenous RD. Expanding research to other RD could shed light on responsive cases warranting SCTA alongside corticosteroid use. Furthermore, longer-term safety studies are required to assess for common AEs known to SCTA use, including cataract progression and IOP elevation.

Several studies have investigated the injection of non-pharmacological substances into the SCS as a medical alternative to scleral buckling procedures for improving outcomes in patients with RD. Gao and colleagues (2019) explored sodium hyaluronate injection into the SCS, followed by retinal hole scleral freezing and laser photocoagulation for rhegmatogenous RD. Remarkably, 50% of the eyes achieved complete reattachment, 33.33% were partially reattached with subsequent reabsorption of subretinal fluid, and 16.67% did not reattach and required further intervention [168]. The concept of sodium hyaluronate acting as an internal buckle through SCS injection was initially explored in rabbit models by Mittl and collaborators (1987). They demonstrated that, while the buckling effect was short-lived (between 12 and 72 h), sodium hyaluronate remained in the SCS for 10 and 14 days, regardless of the concentration or formulation used [169]. Earlier, Smith (1952) reported a series of five RD cases in which air was injected into the SCS to act as an internal buckle. Satisfactory repositioning of the retina was achieved in all five retinas. However, one case experienced vitreous hemorrhage, two cases had relapsed at 2 and 3 months that required surgical management, and one case required further diathermy [170]. The injection of sodium hyaluronate has shown efficacy in the treatment of rhegmatogenous RD in combination with current standards of treatment in one small clinical study. Larger comparative studies are needed to assess if SC sodium hyaluronate results in improved efficacy compared to the current treatment alone. Additionally, further optimization of the formulation of sodium hyaluronate could result in sustained treatment levels that induce buckling via the SCS for a longer duration. While sodium hyaluronate injection has shown promise in small-scale studies, the scarcity of follow-up research since 2019 may be attributed to advancements achieved through SCTA. Moreover, examining the functional and anatomical changes associated with anatomical improvements is crucial. Comparative clinical trials comparing SC sodium hyaluronate and SCTA versus placebo could lead to new treatment regimens that yield better outcomes for patients with various types of RD.

### 7.5. Suprachoroidal Injection for Uveitis

SCTA has demonstrated positive effects on visual and anatomical outcomes with minimal AEs in the treatment of ME secondary to NIU, as previously discussed. Moreover, the literature has provided insights regarding its potential role in managing uveitis. Goldstein and colleagues (2016) were the first to explore the use of 4.0 mg SCTA for eight eyes with NIU to assess the preliminary efficacy and safety of this approach. All treated eyes showed improvements in BCVA by 26 weeks. Among the 38 AEs reported, 89% were mild or moderate in severity, and 58% affected the ocular domain. Notably, 18% were related to uveitis progression, 3% were associated with cataract progression requiring extraction, and 16% were attributed to ocular pain with no cases of increased IOP [171].

Prior to human studies, Noronha and colleagues (2015) examined the use of SCTA in a porcine model of acute uveitis induced by lipopolysaccharide injection. The study assessed the anti-inflammatory effects of SCTA in comparison to oral prednisone. Single eyes of 16 porcine models received either SC salt solution, 2.0 mg SCTA, or 0.1 or 1.0 mg/kg/d of prednisone every 24 h for a total of 3 days, with the other untreated eye acting as a control. The results demonstrated that SCTA (day 1) resulted in more rapid anti-inflammatory effects than oral prednisone (day 3). SCTA was found to be as effective as high-dose oral prednisone and superior to low-dose prednisone. On the first and second days of treatment, SCTA showed lower inflammation scores compared to controls. On the third day, both high-dose prednisone and SCTA had lower inflammation scores than controls. These findings highlighted the advantages of local drug administration with SCTA over systemic prednisone, which has the potential to lead to side effects such as hyperglycemia, immunosuppression, osteoporosis, and adrenal suppression with long-term use [172].

In another study by Gilger and colleagues (2013), the difference between IVTA and SCTA was investigated for the treatment of acute posterior uveitis in a similar porcine model. The researchers compared the efficacy and safety of different doses and administration routes of TA (0.2 mg or 2.0 mg TA using SC or IV injection). The results suggest that 0.2 mg and 2.0 mg of SCTA were equally effective as 2.0 mg IVTA in reducing inflammation and were similar in terms of IOP and OCT measurements. Eyes in the high-dose SCTA group had mean histologic inflammatory scores in the ocular posterior segment that were significantly lower than eyes treated with IVTA, as seen in Figure 7. Additionally, the mean vitreous humor cell count and protein concentration were lower in the high-dose SCTA group when compared to low-dose SCTA and IVTA groups. There were no significant differences in mean aqueous humor protein concentration among the groups, and there were AEs reported within 3 days of treatment [173].

Similarly, Patel and collaborators (2013) conducted a study using a subretinal endotoxin-induced model of panuveitis in rabbits and found that 4.0 mg SCTA was equally effective as IVTA in reducing ocular inflammation. The study lasted for 22 days, and no AEs, including IOP changes, were reported. After 24 h, eyes treated with SCTA showed less panuveitis than IVTA and control eyes. Both SCTA and IVTA resulted in significantly reduced viritis, aqueous flare, cellularity, and histopathological inflammation compared to controls [174].

In 2015, Chen and collaborators used a rabbit model of uveitis induced by lipopolysaccharide to compare the effects of 50 µL (2.0 mg) SCTA and subtenon injection of 20 mg TA. They found that SCTA was well tolerated and provided better therapeutic effects than subtenon 20 mg. Following SCTA, there was an acute elevation in IOP, with higher volumes of SCTA leading to higher IOP. The peak concentration of TA (<1.0 ng/mL) was detected in the retina and posterior vitreous, with nondetectable in the aqueous and 11.6 ng/mL in the plasma. SCTA demonstrated better efficacy with significantly lower aqueous humor cells, lower vitreous opacity scores, and reduced vitreous inflammation on histology when compared to subtenon TA [67].

Porcine models, with similarities in terms of anatomy, size, and retinal vascular pattern to the human eye, have offered valuable insights into the study of SC injection. While animal models are of critical importance, it should be noted that these animal models represent only acute disease and do not fully capture the chronic nature of uveitis, highlighting the need for human clinical trials. Following the improvement in BCVA that was observed in the study by Goldstein and colleagues, further exploration of the use of SCTA for treating ME due to uveitis was conducted by PEACHTREE, as seen previously. However, there still remains a need for these large, long-term, and masked controlled studies to evaluate the efficacy and safety of SCTA for treating the different types of uveitis without ME affecting different parts of the eye.

### 7.6. Suprachoroidal Injection for Glaucoma

Pharmacological treatments for lowering IOP in glaucoma patients often have low bioavailability when administered topically. This results in the need for multiple daily eye drops, leading to poor treatment adherence and systemic side effects [175,176,177,178]. SC injection, which offers higher drug bioavailability at the ciliary body, has gained interest in glaucoma research. Kim and collaborators (2014) found significant dose-sparing of anti-glaucoma medications, Sulprostone (a prostaglandin analog) and Brimonidine (an a2-adrenergic agonist), when injected into the supraciliary zone of the SCS of rabbits compared to topical administration. SC injection of both medications reduced IOP maximally by 3 mmHg in a dose-dependent manner for 9 h [179]. Chiang and colleagues (2016) were able to sustain levels of Brimonidine in the SCS using Brimonidine-loaded poly (lactic acid) microspheres for the treatment of glaucoma for 1 month. In rabbits, these microspheres were found to reduce IOP by 6 mmHg initially and then, by progressively lower amounts for over 1 month. AEs included mild conjunctival redness treated with antibiotic or steroid ointment, difficulty healing at the injection site, and a histological foreign body response to the microspheres with no serious AEs [180].

By employing a direct injection approach into the anterior portion of the SC space, Kim et al. effectively demonstrated therapeutic IOP reduction with lower doses compared to topical treatment in animals. This injection site, termed the supraciliary zone of the SCS, closely neighbors the site of action of many anti-glaucoma drugs, namely the ciliary body. However, prior to initiating human trials to assess the safety and efficacy of the SCS for targeting glaucoma treatment, it is imperative to conduct more extensive safety studies involving more animal models and comparative investigations against current standard treatments. Interestingly, Chiang’s study was able to demonstrate that therapeutic loaded microspheres can also lengthen the time of therapy for up to 1 month. While further refinement of microspheres is needed for optimal results, their study demonstrated that sustained levels of medications in the SCS can be achieved for the treatment of glaucoma for up to 1 month. Notably, high-viscosity formulations injected into this space ensure minimal diffusion toward posterior eye structures, thus enhancing therapeutic effects. These studies collectively underscore the need for additional research aimed at optimizing anti-glaucoma treatment in the supraciliary zone for a longer duration of action using varying drug formulations, viscosities, particle suspensions, particle size, and osmotic and ionic characteristics.

Furthermore, the potential space between the sclera and choroid can temporarily be expanded without long-term AEs and has been well tolerated in animals and humans [66,173,181]. To find a medication and surgery-free method to treat glaucoma, Chae and collaborators (2020) found that SCS expansion using an in situ-forming hydrogel, as seen in Figure 8, reduced IOP in rabbit models for 1 and 4 months, respectively. Their hypothesis was that SCS expansion increases the drainage of aqueous humor from the eye through the uveovortex pathway [182]. No AEs were reported, but minor hemorrhage and fibrosis were observed at the injection site.

Hao and colleagues (2022) confirmed these findings by assessing the effect of an in situ-forming polyzwitterion polycarboxybetaine hydrogel, which decreased IOP for 6 weeks. The treatment was well tolerated with no serious AEs, minimal inflammatory reaction, and histopathological evidence that the SCS became expanded post hydrogel injection [183]. The use of a non-pharmacological approach through SC injection shows promising results in terms of IOP reduction with minor AEs in animal models, such as inflammatory reactions at the injection site. These results pave the way for clinical trials assessing the safety and efficacy of SC expansion to manage glaucoma in humans; however, none have been initiated at present. If shown to be safe and efficacious in human studies, the combination of SC expansion and SC anti-glaucoma medications could stand to lower the incidence of blindness from glaucoma.

### 7.7. Suprachoroidal Injection for Uveal Melanoma

AU-011, also known as Belzupacap Sarotalocan, is a promising treatment for ocular melanoma. This compound triggers cellular necrosis through an immune-mediated response when light is activated. Savinainen and collaborators (2020) found that AU-011 was well distributed in the choroid and resulted in effective anticancer activity due to its long duration of action in rabbit models of choroidal melanoma. Notably, a 100 μL dose of AU-011 remained in the choroid for several days, with distribution across 75% of the posterior globe. Tumor regression and cancer cell necrosis were observed histologically [74]. In 2021, the same researchers determined that SC AU-011 outperformed IV injection in terms of tumor distribution and bioavailability and decreased unintended exposure in a similar rabbit model of choroidal melanoma. Staining showed SC injection resulted in AU-011 tumor penetration at levels five times higher than IV injection, which remained up to 48 h post-injection. In contrast, IV injection resulted in AU-011 staying primarily on the tumor surface. After SC administration, negligible levels of AU-011 were present in the vitreous, and high exposure levels were present in the tumor and choroid–retina [184]. While these two studies focused on AU-011 as a therapeutic in the treatment of choroidal melanoma, Kang and colleagues (2011) found that suprachoroidally injected resin beads and fluorescent microspheres were successfully delivered using a microcatheter to the site of intraocular melanoma. They found no inflammatory reaction associated with the injection [185].

AU-011 stands at the forefront of choroidal melanoma research as an alternative to radiotherapy in preventing vision loss. The interim results of an ongoing multi-center trial for primary choroidal melanoma by IV AU-011 injection have shown the treatment to be well tolerated and produce adequate tumor control, in addition to maintaining vision [186]. Following these results, Demirci and collaborators (2022) demonstrated that SC injection of AU-011 was also safe in the dose escalation phase of the trial. In this phase, 17 subjects with primary indeterminate lesions and small choroidal melanoma received up to three cycles of 3 weekly SC AU-011 injections (max dose of 80 μg) with two rounds of laser. In terms of AEs, 24% experienced anterior chamber inflammation, 12% experienced conjunctival hyperemia, 21% had eye pain, and 12% reported punctate keratitis with no serious AEs related to treatment [187].

While brachytherapy is currently the standard of treatment for uveal melanoma, it is associated with extraocular muscle trauma, radiation toxicity in the form of radiation retinopathy and maculopathy, and ocular conditions such as strabismus, cataracts, and glaucoma [188]. Animal studies have shown that SC injection allows for adequate drug delivery directly to the tumor with a quick onset of action at lower doses. This targeted delivery can increase the range of tumor sizes that can be treated and enable direct penetration while reducing the risk of AEs. The efficacy results of the clinical trial by Demirci and collaborators, to be released later this year, stand to revolutionize the treatment of choroidal melanoma [189]. As ocular AEs can still occur using this method of injection, including changes to BCVA, larger multicenter studies with longer follow-ups are required to assess its safety and to determine long-term remission rates. Future RCTs between patients undergoing radiotherapy, IV AU-011, and SC AU-011 could determine the treatment with the best efficacy and safety profile for patients with uveal melanoma.

### 7.8. Suprachoroidal Injection for Myopia

Currently, there is a lack of animal and clinical studies investigating the use of SC injection for the treatment of myopia. However, Venkatesh and Takkar (2017) proposed that injecting biological cement into the SCS could halt the pathological elongation of the eyeball associated with this condition [190]. In pathological myopia, the sagittal axial length of the eye is longer than expected, which can lead to complications, such as RD [14,191]. However, at present, there is a need for preclinical animal studies to assess safety prior to initiating human trials. While the injection of cement is currently used in a variety of orthopedic conditions, its therapeutic potential in the eye may be limited by the eye’s aqueous environment. Elevating pressure in this area to induce beneficial alterations in elongation mechanics could potentially lead to ocular complications, including trauma. Given the availability of non-invasive, affordable, and effective treatments for myopia, such as glasses and atropine, there might be limited motivation to explore the utilization of SC injection for myopia treatment.

### 7.9. Suprachoroidal Injection for Ocular Inflammatory Diseases

Ketorolac, a short-acting nonsteroidal inflammatory drug, is commonly used topically to relieve ocular inflammation and the resulting pain. However, its topical administration results in suboptimal therapeutic drug levels or AEs with increasing drug dosages. Side effects can include burning and stinging, as well as delayed corneal healing and conjunctival hyperemia [192]. IV injection carries the risk of vitreous opacity and various retinal pathologies [193]. In an animal study involving 54 rabbits, Wang and collaborators (2012) found that IV injection of 250 μg/0.05 mL Ketorolac Tromethamine resulted in higher intraocular concentrations for a longer duration in comparison to SC and IC groups. Mean maximum concentrations of Ketorolac in the vitreous and retina-choroid were highest for IV, followed by the SC and IC injections. In the retina-choroid, there was a statistically significant larger amount of Ketorolac with IV injection compared to SC injection. The half-life of Ketorolac was also longer with IV injection, with plasma concentrations below 0.4 μg/mL in all three groups. Ketorolac remained detectable in the retina-choroid for 24 and 8 h after the IV and SC injection, respectively [194]. Liu and colleagues found unilateral SC injections of 3.0 mg and 6.0 mg Ketorolac Tromethamine in rabbits to be safe as compared to controls in a 2012 animal study. Electroretinography showed no abnormal changes at 1 to 4 weeks post-injection, and the histomorphology of retinal cells was preserved at 4 weeks when compared to the control group [195].

At present, there is limited research on the use of SC Ketorolac in the treatment of ocular inflammatory diseases. Current evidence indicates that up to 6.0 mg of SC Ketorolac was safe from functional and anatomical perspectives on the rabbit retina, warranting larger animal studies to demonstrate efficacy and further safety. Wang and collaborators observed an increase in the maximum mean concentrations of Ketorolac in the vitreous and retina–choroid and an increased half-life for IV as opposed to SC injection. This finding aligns with other CNV animal models showing faster Bevacizumab and Pazopanib clearance with SC injection due to factors that can affect the duration of therapeutic action, such as the drug formulation, volume, viscosity, particle size, in addition to osmotic and ionic characteristics [152,153]. Thus, further studies are required to assess the optimization of Ketorolac drug formulations to obtain a longer duration of action in the SCS. The efficacy and safety of SC injection of Ketorolac should also be compared with other methods of delivery to the eye, including IV injection, retrobulbar injection, and topical administration, to determine the most efficacious approach for patients with a variety of inflammatory diseases, including scleritis, uveitis, keratitis, and conjunctivitis in future larger, multi-center human trials. Interestingly, there have been studies assessing the use of IV injection of Ketorolac, including nonsteroidal anti-inflammatory drugs, as opposed to corticosteroids, in the treatment of CME, DME, CNV, uveitis, and AMD [196,197,198,199]. Due to the low AE profile of nonsteroidal anti-inflammatory drugs compared to corticosteroids, particularly in relation to cataract progression and IOP elevation, further research on its administration using SC injection is warranted.

## 8. Conclusions

Throughout this review, we have endeavored to shed light on the potential of SC injection as an innovative and effective technique for targeted drug delivery to the posterior segment of the eye. Our comprehensive review of the most recent literature has provided compelling evidence of the utility of this minimally invasive method in addressing the challenges associated with traditional treatment approaches. SC injections present a significant advancement over conventional administration routes, such as eye drops and IV injections, offering increased drug bioavailability, extended duration of action, and a marked reduction in off-target adverse effects.

While suprachoroidal injection offers a promising and innovative approach to targeted drug delivery for various ocular conditions, its widespread use is hindered by several current challenges:Biomechanical Considerations: Optimizing the chemical and physical properties of the injectate (such as size and viscosity) is crucial for different indications and diseases, depending on the anatomical location. This requires the refinement of current techniques, including injection speed and approach (single quadrant vs. multiple quadrant), to ensure the precise delivery of the correct amount of drug to the appropriate anatomical site for the desired duration of action;Need for Further Clinical Studies: More phase 3 clinical trials will be essential for broader clinical adoption. As it stands, most studies of indications other than macular edema are either preclinical or early-stage clinical trials. Comprehensive late-stage clinical research is paramount to assess the efficacy, safety, and applicability of suprachoroidal injections across various ocular and retinal conditions;Clinical Translation Challenges: Factors such as drug storage, cost-effectiveness, and efficiency compared to IV injection (the current go-to administration route for posterior segment diseases due to its efficacy) must be carefully considered. Reimbursement considerations also play a vital role in the practical implementation of this technique;Transition Challenges: The shift away from IV to SC injections will not be instantaneous. Strong evidence and concerted efforts will be required for clinicians to be willing to adapt to and learn this new technique and to overcome logistical issues such as in-office procedures.

Despite these challenges, the potential of SC injection is undeniable. Looking forward, it is encouraging to envision a future in which SC injection can be used in conjunction with biotech products, genes, and cell-based therapies to initiate a new era of personalized treatments. This could revolutionize the field of ophthalmology, enhance patient care, and improve outcomes in ocular disease management.

While there is certainly more work to be performed, the path is clear for the continuation of rigorous research into this technique. The potential of SC injection in reshaping the landscape of ocular drug delivery provides a compelling call to action for researchers, clinicians, and stakeholders in the field of ophthalmology. Guided by the powerful intersection of interdisciplinary collaboration, we aim to illuminate a future where devastating vision loss from diseases such as retinitis pigmentosa, severe diabetic retinopathy, and advanced age-related macular degeneration is no longer a life sentence. Together, we strive to restore not only sight but hope, dignity, and quality of life for those grappling with the darkness of these ocular diseases.

## Figures and Tables

**Figure 1 pharmaceuticals-16-01241-f001:**
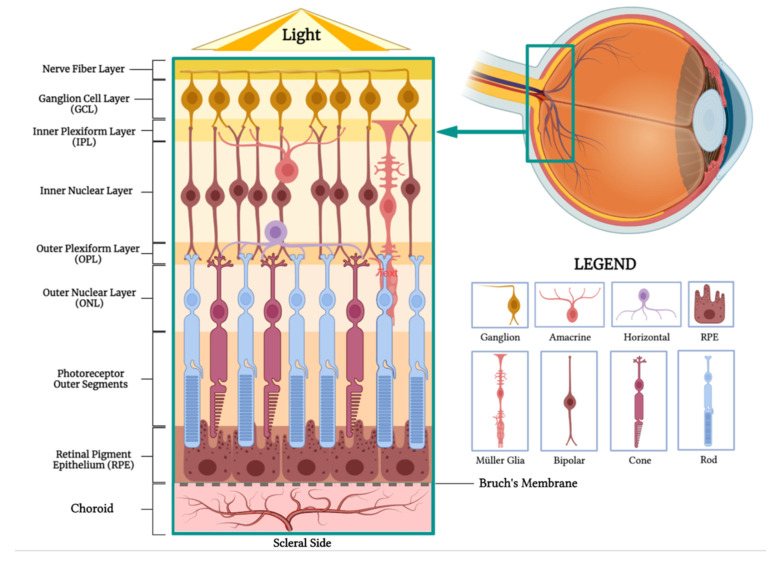
Anatomy of the Retina and Choroid. Critical Role of the Choroid in Nutrient Supply to the Retina.

**Figure 2 pharmaceuticals-16-01241-f002:**
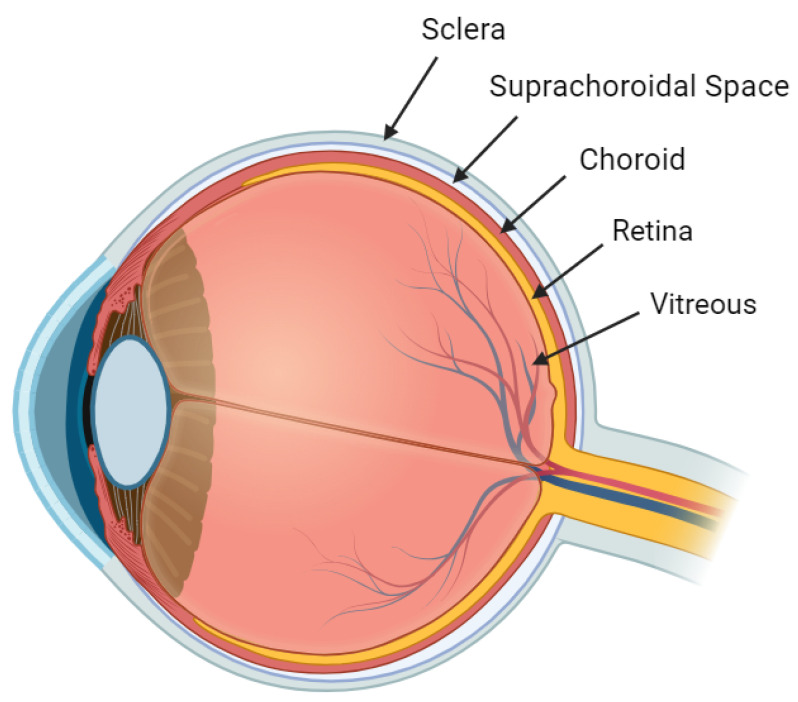
Anatomy of the Suprachoroidal Space and Posterior Segment.

**Figure 3 pharmaceuticals-16-01241-f003:**
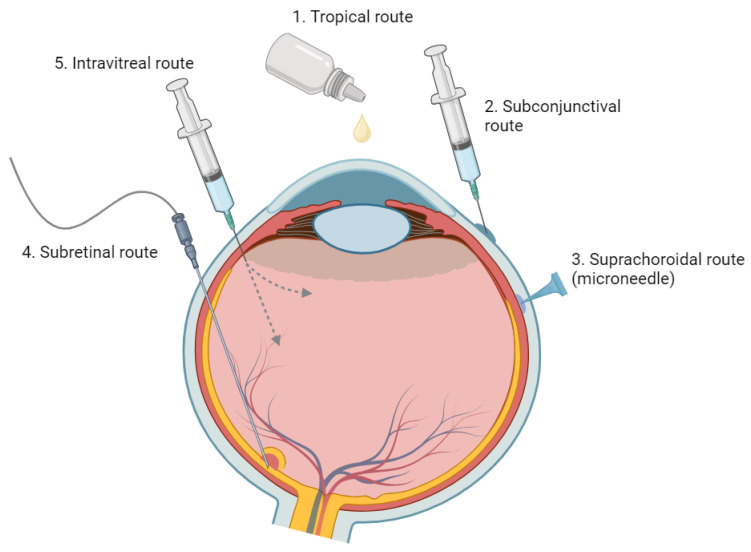
An Overview of Various Ophthalmic Medication Delivery Routes. This figure illustrates the range of administration methods used in ophthalmic medicine, including topical, subconjunctival, intravitreal, suprachoroidal, and subretinal techniques.

**Figure 4 pharmaceuticals-16-01241-f004:**
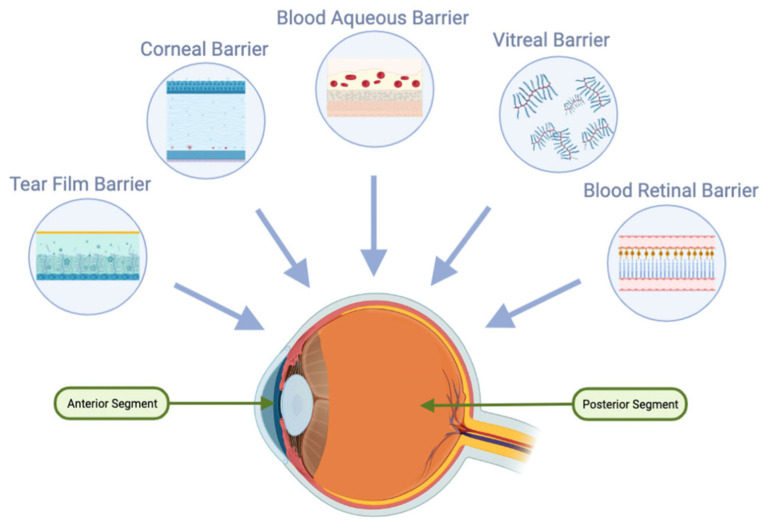
Anatomical and Physiological Barriers in the Eye Impacting Drug Delivery.

**Figure 5 pharmaceuticals-16-01241-f005:**
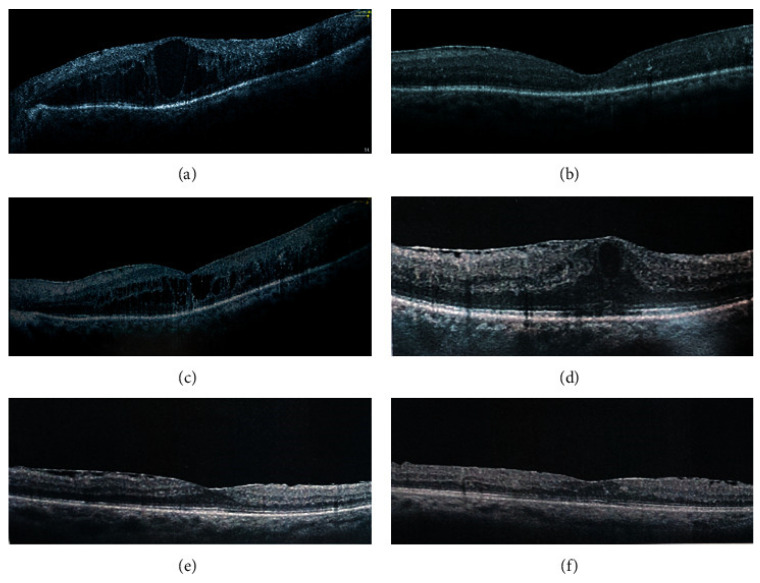
Example of CFT changes measured by OCT after IVTA and SCTA injection at baseline, 1, and 3 months. IVTA group (**a**–**c**). SCTA group (**d**–**f**). Reprinted with permission from Ref. [128]. 2023, Ahmed Abdelshafy Tabl et al.

**Figure 6 pharmaceuticals-16-01241-f006:**
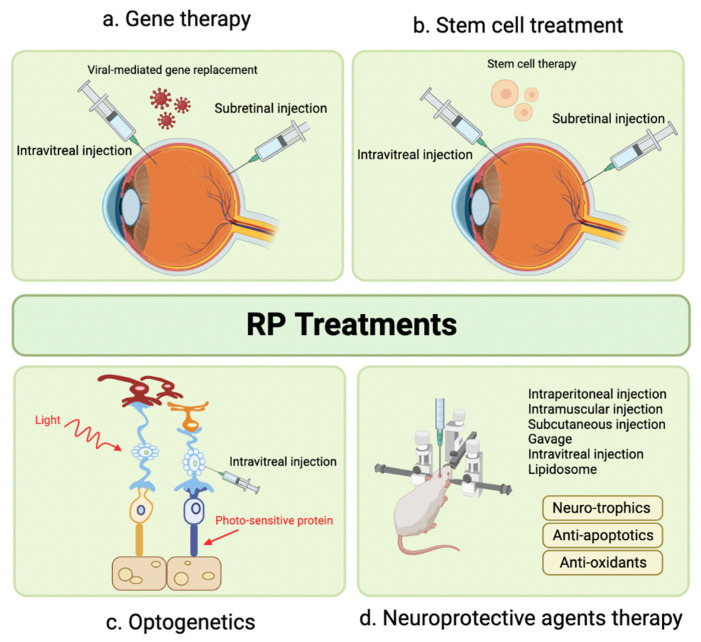
Emerging Therapeutic Modalities for Retinitis Pigmentosa.

**Figure 7 pharmaceuticals-16-01241-f007:**
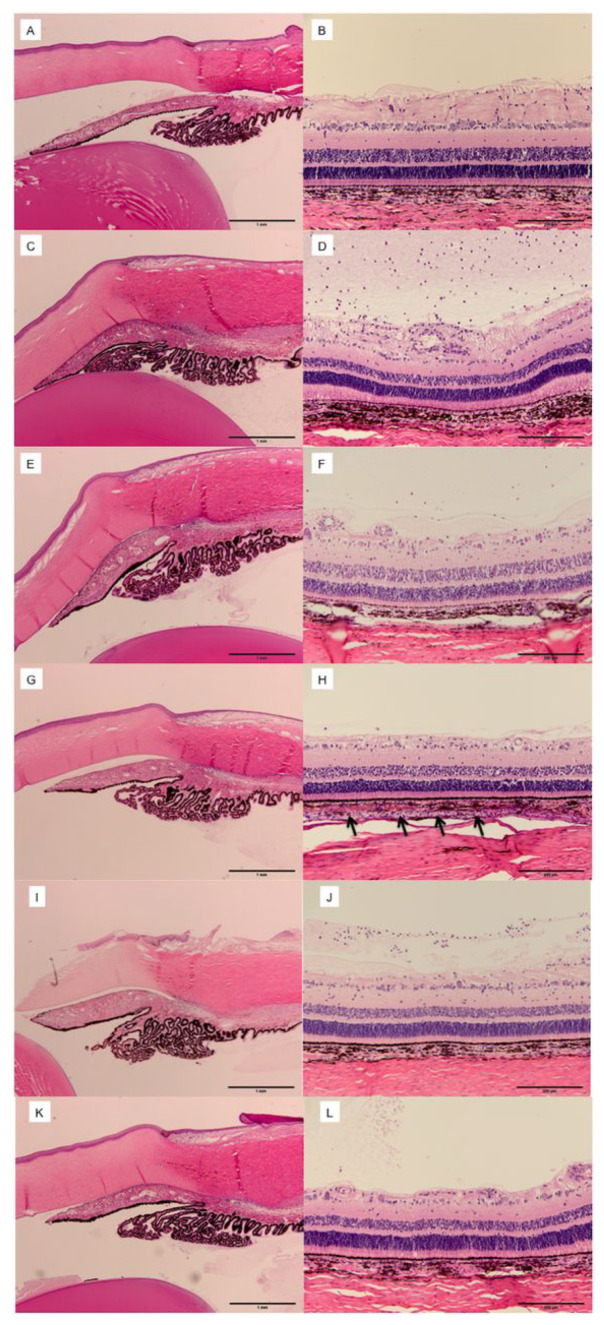
Ocular histopathology of eyes 3 days after IVT injection of balanced salt solution (BSS) or 100 ng of lipopolysaccharide (LPS) and 72 h after SCS or IVT injection of vehicle, 0.2 mg TA (low-dose TA), or 2.0 mg of TA (high-dose TA). Hematoxylin and eosin stain. (**A**) Anterior segment of eyes injected with BSS IVT and vehicle in SCS (group 1). (**B**) Posterior segment of eyes injected with BSS IVT and vehicle in SCS (group 1). (**C**) Anterior segment of eyes injected with LPS IVT and vehicle in SCS (group 2). (**D**) Posterior segment of eyes injected with LPS IVT and vehicle in SCS (group 2). (**E**) Anterior segment of eyes injected with LPS IVT and low-dose TA in SCS (group 3). (**F**) Posterior segment of eyes injected with LPS IVT and low-dose TA in SCS (group 3). (**G**) Anterior segment of eyes injected with LPS IVT and high-dose TA in SCS (group 4). (**H**) Posterior segment of eyes injected with LPS IVT and high-dose TA in SCS (group 4). Arrows indicate presence of TA in SCS. (**I**) Anterior segment of eyes injected with LPS IVT and low-dose TA IVT (group 5). (**J**) Posterior segment of eyes injected with LPS IVT and low-dose TA IVT (group 5). (**K**) Anterior segment of eyes injected with LPS IVT and high-dose TA IVT (group 6). (**L**) Posterior segment of eyes injected with LPS IVT and high-dose TA IVT (group 6). Reproduced with permission [173].

**Figure 8 pharmaceuticals-16-01241-f008:**
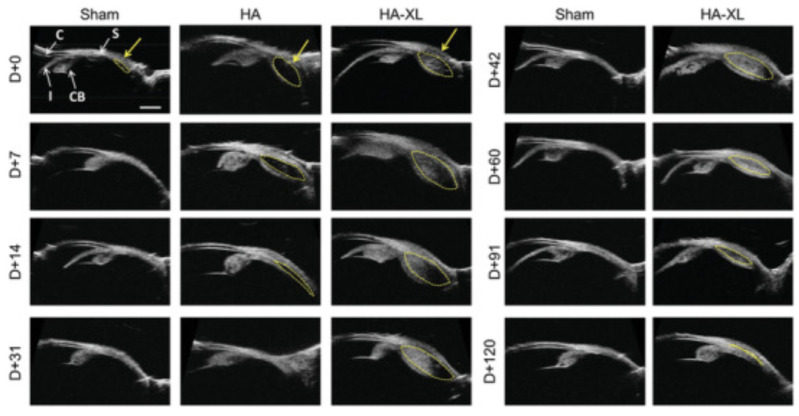
Ultrasound biomicroscopy imaging of hydrogel-injected eyes. Rabbit eyes were injected with Hanks’ Balanced Salt Solution (Sham), commercial hyaluronic acid hydrogel (HA), or in situ-forming hyaluronic acid hydrogel group (HA-XL) and imaged over time. The yellow arrow indicates the approximate injection site, and the yellow dashed line roughly outlines the expanded suprachoroidal space. Images are representative of seven eyes per group (HA-XL group), two eyes per group (Sham), or the only eye available from the HA group. Abbreviations—C: Cornea; CB: Ciliary Body; I: Iris; S: Sclera. D + 0 refers to day zero after injection; D + 7 to 7 days after, etc. Scale bar: 2 mm. Reproduced with permission [182].

**Table 1 pharmaceuticals-16-01241-t001:** Comparison of Different Ocular Drug Administration Methods [2,20,21].

Injection Method	Advantages	Disadvantages
Topical Eye Drops [22]	Prevalent, well-known method	Low bioavailability to posterior segment tissues
	Non-invasive method for ocular drug delivery	Short duration of action, requiring frequent administration
		Relies on patient’s compliance
		Local complications (ocular surface irritation, cataracts, ocular hypertension, periocular aesthetic issues)
Systemic Drug Administration	Noninvasive and potentially patient-preferred	High doses often required due to reduced accessibility to targeted ocular tissues
	Usable as standalone or in combination with topical delivery	Potential systemic side effects due to high dosage, necessitating safety and toxicity considerations
		Effective bioavailability is challenging due to blood–ocular barriers
Intravitreal Injection [23]	Office-based, outpatient procedure	Requires frequent in-office visits
	High bioavailability (bypass corneal and blood–retinal barriers)	Potential for severe complications (Endophthalmitis, retinal detachment, vitreous hemorrhage)
	Fewer systemic side effects compared to oral or IV administration	Local complications (increased IOP, cataract formation)
	Rapid therapeutic onset	Possible post-injection floaters
		Systemic absorption and side effects can still occur
Subretinal Injection [24]	Targeted treatment for the RPE and outer retina	Invasive procedure, requires vitrectomy
	Reduced immune reactions for gene therapy using viral vectors (due to injection in an immune-privileged site)	Limited distribution of injectate within subretinal space; effects confined to injection site

## Data Availability

Data sharing is not applicable.

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
