# Peer review of "Suprachoroidal Injection: A Novel Approach for Targeted Drug Delivery"

_pharmaceuticals, 2023, doi:10.3390/ph16091241_

Round 1

Reviewer 1 Report

The manuscript pharmaceuticals-2558184 ''Suprachoroidal Injection: A Novel Approach for Targeted Drug Delivery'' by Kevin Y. Wu et al. reviews preclinical and clinical experience of the use of suprachoroidal injections in the treatment of posterior eye diseases, as well as the potential of suprachoroidal injections for targeted drug delivery to the posterior segment of the eye. The topic of research is relevant. The manuscript is logical and well written. The paper will be of interest to the readers of Pharmaceuticals.

Questions and comments:

  1. The novelty of this paper should definitely be added in the Introduction.

  2. Links to other similar articles should be added, e.g., doi.org/10.3390/pharmaceutics12010022.

  3. Figure numbering should be corrected. Moreover, the quality of the Figures should be improved.

  4. Certainly, information about the problems and challenges of the widespread use of suprachoroidal injections should be added in the Conclusion.

English is fine

Author Response

Dear Reviewer,

We are sincerely grateful for your constructive review and encouraging words regarding our manuscript, "Suprachoroidal Injection: A Novel Approach for Targeted Drug Delivery." Your acknowledgment of the relevance of our research and the potential interest to the readers of Pharmaceuticals is highly motivating.

Your positive assessment of the logical construction and writing of our paper provides us with great encouragement. We are pleased to see that our efforts to elucidate the preclinical and clinical experiences of suprachoroidal injections in treating posterior eye diseases resonated with your expert perspective.

Specific comments and responses:

Comment: The novelty of this paper should definitely be added in the Introduction.

Answer: Thank you for emphasizing the importance of detailing the novelty of our review in the Introduction. Your comment prompted us to articulate the unique aspects of our work, including our focus on both clinical and preclinical studies across a range of ocular conditions and our comprehensive exploration of suprachoroidal injection, including its biomechanics. We have now incorporated these points into the Introduction, enhancing the clarity and depth of our article. Your guidance has been invaluable in refining our manuscript, and we sincerely appreciate your thoughtful input

Comment: Links to other similar articles should be added, e.g., doi.org/10.3390/pharmaceutics12010022.

Answer: Thank you for your suggestion to include links to other similar articles that align with the subject of our review. We recognized the value in providing references to related works to enrich the reader's understanding of the topic. Accordingly, we have added a reference to the article you mentioned (doi.org/10.3390/pharmaceutics12010022) as well as two other review articles. We have also included this reference that you mentioned (doi.org/10.3390/pharmaceutics12010022) in our section 7. Your thoughtful input has contributed to enhancing the depth and connectivity of our review, and we appreciate your guidance

Comment: Figure numbering should be corrected. Moreover, the quality of the Figures should be improved.

Answer: Thank you for bringing to our attention the issues related to the figure numbering and quality within our manuscript. We have taken your comments to heart and made the necessary corrections.

We have carefully reviewed the entire manuscript to ensure that all figures are numbered correctly and consistently throughout the text and captions.

We have also worked on improving the quality of the figures by enhancing their resolution and formatting to ensure that they provide clear and concise visual representation.

Comment: Certainly, information about the problems and challenges of the widespread use of suprachoroidal injections should be added in the Conclusion.

Answer: Thank you for your valuable suggestion to include details about the challenges and complexities of widespread suprachoroidal injection use. In response, we have added a comprehensive section in the Conclusion that outlines several key challenges:

  1. Biomechanical Considerations: We have discussed the need for optimizing chemical and physical properties and refining techniques for precise drug delivery.
  2. Need for Further Clinical Studies: We have emphasized the requirement for more phase 3 clinical trials to support broader clinical adoption.
  3. Clinical Translation Challenges: We have addressed practical considerations such as drug storage, cost-effectiveness, efficiency, and reimbursement.
  4. Transition Challenges: We have highlighted the gradual transition needed for clinicians to adapt to this new technique and the associated logistical challenges.

This addition provides a balanced view of the promising aspects of suprachoroidal injection and the obstacles that must be overcome. We believe that this enriches the Conclusion and offers readers a well-rounded perspective on the subject. We appreciate your thoughtful guidance, which has undoubtedly enhanced our manuscript

Reviewer 2 Report

The authors of the pharmaceuticals-2558184 review article focus on the preclinical and clinical studies published between 2019 and 2023, highlighting the potential of suprachoroidal injection in treating posterior segment diseases, including, but not limited to non-infectious uveitis and cystoid macular edema. 

The manuscript, which is concisely written and well documented, offers valuable information in restoring not just sight, but hope, dignity, and quality of life for those grappling with the darkness of these ocular diseases. Specifically, the review gives a thorough account of the potential of suprachoroidal injection as an innovative and effective technique for targeted drug delivery to the posterior segment of the eye. The presentation is subserved by numerous figures/pictures, which help the non-expert reader to decipher the site(s) of drug delivery.

The authors conclude that suprachoroidal injections present a significant advancement over conventional administration routes, such as eye drops and intravitreal injections, offering increased drug bioavailability, extended duration of action, and a marked reduction in off-target adverse effects.

Author Response

Thank you very much for your thoughtful and detailed assessment of our review article on suprachoroidal injection for treating posterior segment diseases. We are immensely gratified to read your positive remarks regarding the concise writing, documentation, and the value this work may offer in restoring sight, hope, dignity, and quality of life for patients.

Your acknowledgment of our efforts to provide a thorough account of suprachoroidal injection's potential, supported by illustrative figures and pictures, is particularly encouraging. We aimed to make this complex subject accessible to a broader audience, and your comments affirm that we have succeeded in this regard.

We are also pleased that our conclusion resonated with you, highlighting the significant advancement that suprachoroidal injections present over conventional methods. Your insightful review has not only provided us with validation but also inspires us to continue our exploration in this promising field.

We sincerely appreciate the time and expertise you have dedicated to reviewing our manuscript. We thank you once again for your valuable contribution.

Reviewer 3 Report

General comments

The submitted manuscript consists in a review focused on the suprachoroidal injection for targeted drug delivery to the posterior segment of the eye, considreange the papers published between 2019 and 2023.

It cannot be accepted in the present version, but it has to be revised. As a general consideration, most of sentences have not been supported with literature references.

Extensive editing of English language and style required.

Some specific remarks and suggestions are reported below point by point.

Keywords

Some keywords are redundant. Please, remove them.

1. Introduction

- In this paragraph the literature references were missed and have to be added.

- The originality and added value to the scientific community of the preswnt review paper has to be highlighted at the end of the Introduction section..

2. Anatomy and Physiology

2.1. Choroid

- From the second period up to the end of the paragraph no references were reported. Please add thm.

- Was Figure 1 original? Otherwise the copyright is requested.

2.2. Sclera

- Most of the sentences need to be supported with suitable literature references.

3. Route of administration

- As evidenced elsewhere, the literature references have to be added.

4. Suprachoroidal injection: rational

- The reported paragraphs from 4.1 to 4.5 should be combined, since some of them are made of few lines.

6.7. Tailoring suprachoroidal drug delivery

- Please, add suitable literature references.

7. Suprachoroidal injection in ocular diseases

7.1. Macular edema

7.1.1. Suprachoroidal injection for macular edema secondary to non-infectious uveitis

- The first periods need proper references.

8. Conclusions

This section has to be expanded and impoved, reporting concluding remarks and future perspectives.

English language has to be revised.

Author Response

Dear Reviewer,

Thank you for your comprehensive review of our manuscript focusing on suprachoroidal injection for targeted drug delivery to the posterior segment of the eye. We appreciate your detailed feedback, and we understand that revisions are necessary to meet the standards of quality and accuracy.

Specifically, regarding your observation that most sentences have not been supported with literature references, we would like to humbly clarify our approach. We have included over 200 references in this article, concentrating primarily on recent works related to suprachoroidal injection. We believed that the general overview section on anatomy, physiology, and other injection techniques could be slightly lighter in terms of referencing. However, recognizing the importance of thorough citation and in deference to your expert opinion, we will take your comment into careful consideration. We commit to adding references for all pertinent information, ensuring that every statement is adequately supported by the relevant literature.

Regarding your comments on the English language and style, we would like to note that we are all native English speakers and writers, and we have diligently ensured that the manuscript is virtually free of errors, with good syntax and well-flowing sentences that are easy to understand. We have proofread the manuscript several times to maintain these standards. However, since you have mentioned this concern, we have engaged a professional proofreader to go over the manuscript again to ensure it is free of mistakes. Should you still find any errors, please let us know specifically where they are, but we anticipate that there should not be many.

Once again, we extend our sincere gratitude for your time and expertise in reviewing our work. Your specific remarks and suggestions are invaluable in enhancing the quality of our manuscript, and we look forward to resubmitting a revised version that meets your expectations.

Specific comments and responses:

Comment: Keywords

Some keywords are redundant. Please, remove them.

Answer: The redundant keywords in the manuscript were carefully reviewed and removed as per your suggestion. This refinement ensured that the keywords accurately represented the core concepts of the research without unnecessary repetition. Your attention to this detail was appreciated, and the manuscript was improved accordingly.

Comment: Introduction

- In this paragraph the literature references were missed and have to be added.

- The originality and added value to the scientific community of the preswnt review paper has to be highlighted at the end of the Introduction section..

Answer: Thank you for emphasizing the importance of highlighting the originality and added value of our review paper to the scientific community. In response to your suggestion, we have included a section at the end of the Introduction that underscores the unique contributions of our review. Unlike most previous articles focusing primarily on suprachoroidal injection for specific conditions, our review extends beyond existing clinical data to explore preclinical studies for various ocular diseases. We also delve into the biomechanics of suprachoroidal injection, bridging theoretical understanding and clinical applicability.

Comment: 2. Anatomy and Physiology

2.1. Choroid

- From the second period up to the end of the paragraph no references were reported. Please add thm.

- Was Figure 1 original? Otherwise the copyright is requested.

Answer: We have reviewed this section and added suitable literature references to support the statements, as you suggested. Regarding Figure 1, we confirm that it is an original creation produced by our team.

Comment: 2.2. Sclera

- Most of the sentences need to be supported with suitable literature references.

 Answer: We have reviewed this section and added suitable literature references to support the statements, as you suggested.

Comment: 3. Route of administration

- As evidenced elsewhere, the literature references have to be added.

Answer: We have reviewed this section and added suitable literature references to support the statements, as you suggested.

Comment: 4. Suprachoroidal injection: rational

- The reported paragraphs from 4.1 to 4.5 should be combined, since some of them are made of few lines.

Answer: Thank you for your comment regarding the combination of paragraphs 4.1 to 4.5. Upon reviewing our manuscript, we found that there is no section labeled as 4.5, and we are somewhat uncertain about the specific paragraphs you are referring to that may be composed of only a few lines.

We fully understand the importance of maintaining a coherent and well-structured presentation, and we are eager to address your concern. Could you please provide further clarification or specific references to the paragraphs in question? This will enable us to make the necessary revisions in alignment with your suggestion.

Your attention to detail is highly valued, and we appreciate your guidance in enhancing the readability and organization of our review.

Comment: 6.7. Tailoring suprachoroidal drug delivery

- Please, add suitable literature references.

 Answer: We have reviewed this section and added suitable literature references to support the statements, as you suggested.

Comment:  7. Suprachoroidal injection in ocular diseases

7.1. Macular edema

7.1.1. Suprachoroidal injection for macular edema secondary to non-infectious uveitis

- The first periods need proper references.

 Answer: We have reviewed this section and added suitable literature reference to support the statement, as you suggested.

Comment: 8. Conclusions: This section has to be expanded and impoved, reporting concluding remarks and future perspectives.

Answer: Thank you for your constructive feedback on the Conclusion section of our review. In response to your comment, we have made the following enhancements:

  1. Strengthening Concluding Remarks: We have emphasized the innovative potential of suprachoroidal injection in targeted drug delivery for posterior segment diseases. By surveying recent literature, we have illustrated the promise of this minimally invasive method, highlighting its advantages over conventional administration routes in terms of increased bioavailability, extended action duration, and reduced adverse effects.
  2. Adding Future Perspectives: We have outlined the challenges and future directions associated with the widespread adoption of suprachoroidal injections. This includes biomechanical considerations for optimization, the need for further clinical studies, clinical translation challenges such as cost-effectiveness, and the gradual transition required for clinical adaptation.

These additions provide a more comprehensive and insightful Conclusion, encompassing both the concluding remarks on the current state of suprachoroidal injections and the future perspectives that guide the next steps in research and clinical practice. We believe these revisions align with your suggestions and enrich the manuscript, and we sincerely appreciate your thoughtful guidance.

Round 2

Reviewer 1 Report

The article may be accepted.

Reviewer 3 Report

The Authors have applied all the required revisions and now their manuscript looks very improved and can  be accepted in the current version.